# Sea lamprey enlightens the origin of the coupling of retinoic acid signaling to vertebrate hindbrain segmentation

Alice M. H. Bedois [1], Hugo J. Parker[1], Andrew J. Price [1], Jason A. Morrison [1], Marianne E. Bronner[2] & Robb Krumlauf [1,3] ✉

Retinoic acid (RA) is involved in antero-posterior patterning of the chordate body axis and, in jawed vertebrates, has been shown to play a major role at multiple levels of the gene regulatory network (GRN) regulating hindbrain segmentation. Knowing when and how RA became coupled to the core hindbrain GRN is important for understanding how ancient signaling pathways and patterning genes can evolve and generate diversity. Hence, we investigated the link between RA signaling and hindbrain segmentation in the sea lamprey *Petromyzon marinus*, an important jawless vertebrate model providing clues to decipher ancestral vertebrate features. Combining genomics, gene expression, and functional analyses of major components involved in RA synthesis (Aldh1as) and degradation (Cyp26s), we demonstrate that RA signaling is coupled to hindbrain segmentation in lamprey. Thus, the link between RA signaling and hindbrain segmentation is a pan vertebrate feature of the hindbrain and likely evolved at the base of vertebrates.

The body plan of most deuterostomes develops using a similar set of transcription factors (TFs) and signaling pathways (e.g., FGF, Wnt) which are expressed in analogous axial domains of different developing embryos[1–5]. This suggests the presence of an ancient, conserved core gene regulatory network (GRN) underlying axial patterning, which integrates inputs from developmental TFs and signaling pathways. For example, in the evolution of chordates the coordinated action of FGF and Wnt signaling[2,6,7] coupled with the Hox family of TFs is essential for establishing and patterning the antero-posterior (A-P) axis[5,8–10].

In chordate embryos, nested domains of *Hox* expression play a fundamental role in regulating patterning of the nervous system[1,11–14]. In addition to inputs from FGF and Wnt signaling, evidence from cephalochordate and jawed vertebrate models indicate that retinoic acid (RA) signaling plays a key role in coordinating the regulation of *Hox* gene expression along the A-P axis[5,7,15–23]. Regulatory studies have also uncovered the presence of retinoic acid response elements (RAREs) present in conserved positions in an amphioxus and mouse

*Hox* cluster that contribute to nested domains of *Hox* expression[24–26]. This suggests that a direct regulatory link between RA signaling and *Hox* expression may be an ancient feature of the core GRN coupled to A-P patterning of the body axis in all chordates (Fig. 1a)[26].

In jawed vertebrates this RA/*Hox* regulatory hierarchy is coupled to hindbrain segmentation, a vertebrate specific feature of the nervous system (Fig. 1a). The hindbrain is a complex coordination center that regulates vital functions and behaviors and is the site of origin of a subset of neural crest cells, whose derivatives form many craniofacial structures[14,27–30]. Hence, processes that form and pattern the hindbrain are believed to be important for the evolution and emergence of a diversified and complex head in vertebrates[31–33].

During early embryogenesis, the hindbrain is transiently organized into segments (rhombomeres), which lays down a ground plan for regional patterning of neural differentiation, circuit formation and head development[14,26,34]. Hindbrain segmentation is regulated by a conserved GRN which can be visualized as a hierarchical series of regulatory modules that govern sequential steps of the cellular and

[1]Stowers Institute for Medical Research, Kansas City, MO 64110, USA. [2]Division of Biology and Biological Engineering, California Institute of Technology, Pasadena, CA 91125, USA. [3]Department of Anatomy and Cell Biology, Kansas University Medical Center, Kansas City, MO 66160, USA.
✉e-mail: rek@stowers.org

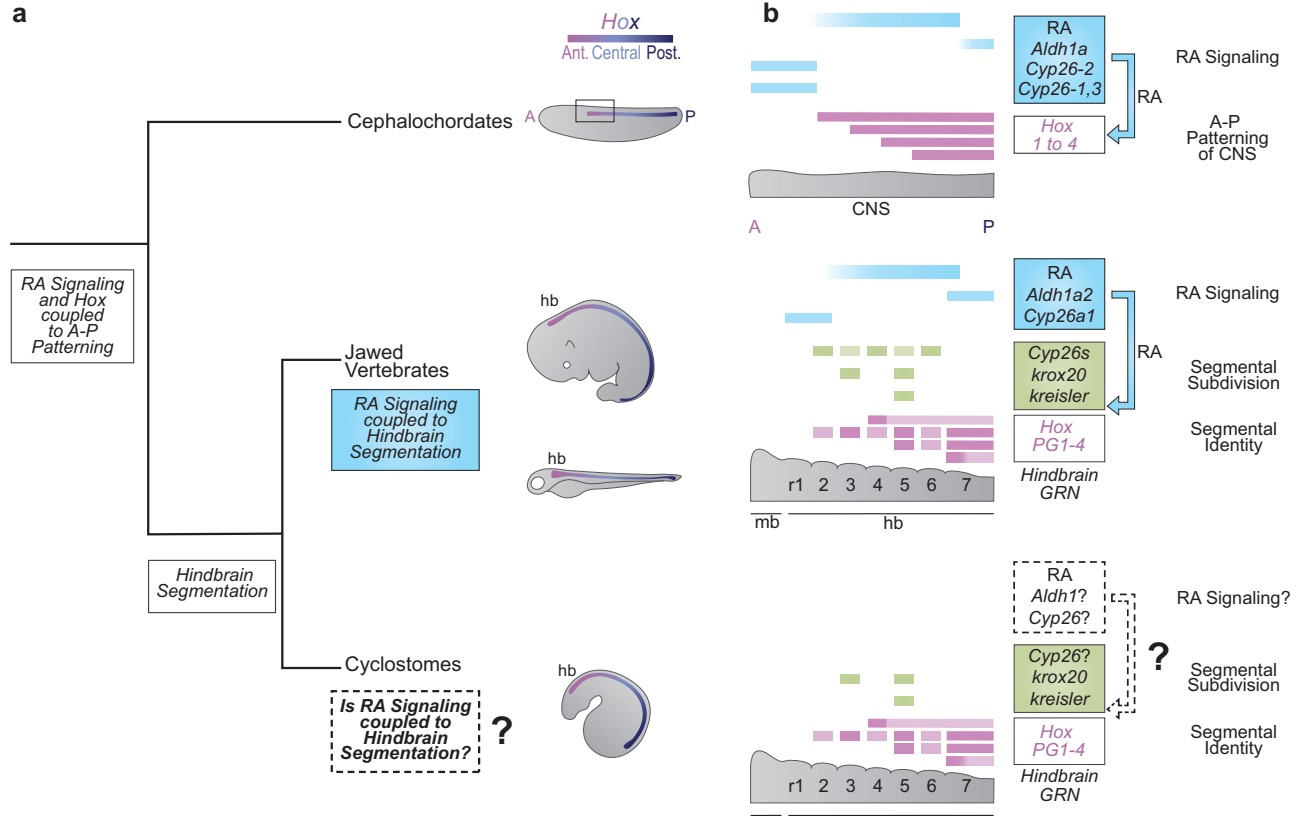

**Fig. 1 | What is the origin of the coupling of RA signaling to hindbrain segmentation? a** An ancient RA/*Hox* hierarchy is involved in axial (A-P) patterning in all chordates (e.g., *Aldh1as, Cyp26s, Hox 1-4*); **b** Many of these ancestral genes are wired into a complex and dynamic GRN underlying the process of hindbrain segmentation in jawed vertebrates. Some aspects of the GRN have been shown to be highly conserved in sea lamprey (e.g., segmental expression of *Hox*

*PG1-4, krox20* and *kreisler*) suggesting that hindbrain segmentation originated early in vertebrate evolution. However, it remains unknown whether RA signaling plays a role in the GRN for segmentation in sea lamprey. Here we use the sea lamprey as a jawless vertebrate model to understand the origin of the coupling of RA to the hindbrain GRN for segmentation in the evolution of vertebrates.

developmental patterning process. The GRN is initiated by signaling cues, including RA, which setup TFs that subdivide the region into segments, followed by a patterning module, which imparts unique properties to each segment (Fig. 1b)[14,19,35,36]. In jawed vertebrates, RA signaling is essential for the initiation and regulation of multiple modules, and *Hox* genes subsequently play important functions in specifying the identity of segments (Fig. 1b). The importance of hindbrain segmentation for craniofacial development of jawed vertebrates raises the question of when during vertebrate evolution the ancient RA/*Hox* hierarchy became coupled to the process of hindbrain segmentation. Exploring the origin of the coupling of RA signaling to hindbrain segmentation is also important for understanding how ancient signaling pathways and patterning genes can evolve their regulatory interactions to generate more complex GRNs that contribute to morphological diversity.

Lampreys and hagfish belong to a group of jawless vertebrates (cyclostomes) that diverged from other vertebrates ~500 million years ago[37]. Thus, they have a unique position in the vertebrate tree as a sister group to jawed vertebrates and constitute important models for understanding the evolution of vertebrate traits[38]. It was previously postulated that lamprey hindbrain segmentation is only partially coupled to *Hox* expression, and that RA signaling influences *Hox*-dependent branchiomotor neuron specification but not hindbrain segmentation itself[39,40]. This implies that roles for RA in hindbrain segmentation may have arisen later in vertebrate evolution. This finding is consistent with the idea that there has been a gain in connectivity between GRNs and signaling pathways in the evolution of

jawed vertebrates from their chordate ancestors, which may have contributed to the formation of new cell types and morphological novelties[41]. However, through gene expression and cross-species regulatory studies we have recently uncovered clear evidence that *Hox* genes and other TFs of the hindbrain GRN in jawed vertebrates are also coupled to the process of segmentation in the sea lamprey (*Petromyzon marinus*) (Fig. 1b)[42,43]. This conservation implies that hindbrain segmentation is a pan-vertebrate trait. Paradoxically, RA signaling is not thought to be involved in segmental processes in lamprey[39,40], while it plays major roles in regulating hindbrain segmentation in jawed vertebrates. This disparity raises important questions about the origin of the role of RA signaling in hindbrain segmentation and the evolution of the vertebrate hindbrain GRN.

In light of our findings regarding the presence of key regulatory components of the hindbrain GRN in lamprey[42,43], we wanted to re-examine whether or not RA is coupled to this GRN. Thus, we investigated the link between RA signaling and hindbrain segmentation using sea lamprey as a jawless vertebrate model, comparing it to the current knowledge built from other jawed vertebrates. Combining genomic, gene expression, and functional analyses of major components involved in the synthesis (Aldh1a enzymes) and degradation (Cyp26 enzymes) of RA, we demonstrate that RA signaling is coupled to hindbrain segmentation in lamprey. Our findings reveal that the GRN for hindbrain segmentation and the roles for RA signaling in its regulation were already present in the vertebrate ancestor before the split between jawless and jawed vertebrates.

## Results

### Analysis of sea lamprey *Cyp26* and *Aldh1a* gene families

To investigate potential roles for RA signaling in hindbrain development of the sea lamprey (*Petromyzon marinus, Pm*), we first identified members of gene families predicted to encode components of the enzymatic machinery involved in synthesis (Aldh1a) and degradation (Cyp26) of retinoids associated with neural tube development in jawed vertebrates[26]. We identified three predicted *Cyp26-like* genes in the sea lamprey germline genome assembly (KPetmar1)[44,45], *Cyp26A1*, *Cyp26B1/C1a*, and *Cyp26B1/C1b*, and two *Aldh1a-like* genes, *Aldh1a1/a2a* and *Aldh1a1/a2b*. The lamprey *Aldh1a1/a2a* gene corresponds to the previously identified *Aldh1a2* gene[46]. Within the cyclostome group, we also searched the hagfish genome (Eburgeri3.2) and found two *Cyp26-like* genes and one *Aldh1a1* gene. The gene structures and lengths of the predicted proteins of lamprey and hagfish *Cyp26* and *Aldh1a* genes show a high degree of similarity to their putative jawed vertebrate counterpart genes (Supplementary Tables 1–4).

To explore properties of the enzymes encoded by the lamprey *Cyp26* and *Aldh1a* genes, we searched for specific catalytic sites essential for the activity of these protein families in other vertebrate species. Protein alignments of the Cyp26 family show conserved amino-acid (AA) sequences for the I/K Helices and the Heme domain in lamprey and hagfish (Supplementary Fig. 1)[47]. Similarly, alignments between Aldh1a proteins reveal that key catalytic domains of this family - the Cysteine (Cys) and Glutamic Acid (Glu) residues - are conserved in the lamprey Aldh1a1/a2 proteins (Supplementary Fig. 2)[48].

Phylogenetic analyses of predicted protein sequences were performed to examine their evolutionary relationship with jawed vertebrate *Aldh1a* and *Cyp26* genes. All vertebrate *Cyp26A1* genes group together as a clade, with high bootstrap support (81) (Fig. 2a). We did not retrieve a *Cyp26A1* homolog in the hagfish genome, which may reflect an incomplete assembly or that *Cyp26A1* was lost in hagfish. The vertebrate *Cyp26B1/C1* genes form a sister clade to the *Cyp26A1* clade, with strong bootstrap support (99) (Fig. 2a), indicating a separation between *Cyp26A1* and *Cyp26B1/C1* genes prior to the divergence of jawed and jawless vertebrates. Within the *Cyp26B1/C1* clade, there is support for separate jawed vertebrate *Cyp26C1* and *Cyp26B1* clades. While lamprey *Cyp26B1/C1a* and *Cyp26B1/C1b* appear to group with the two hagfish *Cyp26C1-like* genes, we are unable to infer any 1:1 orthology between cyclostomes and jawed vertebrates. In addition, hidden paralogy between cyclostome and gnathostome Cyp26A1s cannot be excluded based on the phylogenetic analysis. Thus, it is possible that the cyclostome and gnathostome Cyp26A1s may not be direct orthologues.

For the *Aldh1a* complement, the tree shows clear jawed vertebrate clades for *Aldh1a1* and *Aldh1a2*. However, it does not resolve clear relationships between the cyclostome and jawed vertebrate *Aldh1a* genes, since the cyclostome *Aldh1a* genes do not group with either *Aldh1a1* or *Aldh1a2* clades, nor with each other. In jawed vertebrates, the lineage that led to *Aldh1a3* genes appears to have diverged early from the ancestor that led to the *Aldh1a1/Aldh1a2* clade[49]. Hence, we examined whether the *Pm Aldh1a1/a2* genes could be related to *Aldh1a3*. However, including the *Aldh1a3* family in this analysis did not help to resolve the relationship of the sea lamprey *Aldh1a1/a2a* and *Aldh1a1/a2b* with the vertebrate *Aldh1as* (Supplementary Fig. 3).

### Synteny analysis of vertebrate *Cyp26* and *Aldh1a* genomic loci

To gain further insight into the relationship between the sea lamprey and jawed vertebrate *Cyp26* and *Aldh1a* complements, we looked for evidence of conserved local synteny across vertebrate lineages (Fig. 2b). Despite observing clear syntenic groups in the jawed vertebrate *Aldh1a1* and *Aldh1a2* genomic loci, we did not find any indication of shared synteny with lamprey *Aldh1as* (Fig. 2b).

In most jawed vertebrate lineages *Cyp26A1* is positioned directly adjacent to a *Cyp26C1* gene (Fig. 2b). In sea lamprey, *Pm Cyp26A1* and

*Cyp26B1/C1a* are also located adjacent to each other on chromosome 11, and we uncovered evidence of syntenic relationships around the *Pm Cyp26A1-Cyp26B1/C1a* locus (*Myof, Exo6*) shared with other vertebrates. These similarities in the genomic organization suggest that these genes arose by tandem duplication before the split between jawed and jawless vertebrates, and that this arrangement was subsequently maintained in most vertebrate lineages. Our analysis of the third *Cyp26* gene, *Cyp26B1* or lamprey *Cyp26B1/C1b*, also reveals evidence for shared synteny (*Dysf* and *Exo6b*) between lamprey and jawed vertebrates (Fig. 2b).

In summary, the data from phylogenetic and synteny analyses support an early separation of vertebrate *Cyp26A1* and *Cyp26B1/C1* genes prior to the divergence of jawed and jawless vertebrate lineages. Lamprey *Cyp26A1* appears to be a clear ortholog of jawed vertebrate *Cyp26A1*, but the evolutionary history of the *Cyp26B1/C1* group in vertebrates is less clear. This suggests that an ancestral vertebrate had one *Cyp26A1* gene and at least one *Cyp26B1/C1-like* gene. In contrast, the ancestry of the vertebrate *Aldh1a* genes is not as clear, with cyclostome genes forming basal branches in the phylogeny and not falling within either of the jawed vertebrate *Aldh1a1* and *Aldh1a2* clades.

### Expression analysis of sea lamprey *Cyp26* and *Aldh1a* genes

We next investigated the expression patterns of *Cyp26* and *Aldh1a* genes in sea lamprey embryos. We analyzed existing RNAseq data sets for a series of early developmental stages[50,51] to determine whether any of these genes are expressed during periods relevant to hindbrain segmentation (Fig. 3a). *Cyp26A1* and *Aldh1a1/a2b* are expressed during blastula and gastrulation stages (Tahara stages st7-st17)[52] but are downregulated during neurulation and hindbrain segmentation (st17-st24). In contrast, *Cyp26B1/C1a* and *Aldh1a1/a2a* are not expressed at the earliest stages but are upregulated during gastrulation, neurulation, and segmentation (Fig. 3a). *Cyp26B1/C1b* is not expressed at any of these developmental times.

Based on these data, we focused on *Cyp26A1, Cyp26B1/C1a* and *Aldh1a1/a2a* and characterized their spatio-temporal expression patterns in the period between mid-gastrulation and late hindbrain segmentation (st16-st23) by colorimetric in situ hybridization (cISH) (Fig. 3b, c). *Cyp26A1* expression is dynamic over these stages. Expression is detected in the anterior neural plate at the beginning of neurulation (st16), persists between st17-st19 and then is progressively down-regulated in the most anterior part of the developing neural tube (Fig. 3b). At st20 and later stages, low levels of *Cyp26A1* expression are detected in lateral regions of the head, which appear to correspond to the optic placode, and in the neural tube at the level of the caudal hindbrain.

For *Cyp26B1/C1a*, between st16-st20, weak expression is observed in small patches lateral to the developing neural tube and at st20 a restricted domain of expression is visible within the hindbrain primordium (Fig. 3b). Between st21-st23, two non-adjacent stripes are observed in the hindbrain and then form a series of stripes of differing intensities that cover most of the hindbrain. This dynamic pattern is reminiscent of the expression of *Cyp26B1* and *Cyp26C1* in specific rhombomeres of zebrafish and mouse embryos[36,53,54], suggesting that *Cyp26B1/C1a* could also be coupled to segmentation in lamprey.

A previous study characterized the expression of *Aldh1a1/a2a* in dorsal interneurons of the lamprey spinal cord (st22-st25) and linked this pattern with an ancient conserved intronic enhancer[46]. In earlier stages, we find that expression of *Aldh1a1/a2a* is first visible at st16 around the blastopore. At st17, during the early stages of neurulation, it is highly expressed in two posterior domains corresponding to the presomitic mesoderm (PSM) and this expression is maintained as the embryo undergoes neurulation (Fig. 3c). At st19/20, *Aldh1a1/a2a* is expressed in the PSM and newly developing somites adjacent to the neural tube, as well as in the lateral plate mesoderm in the posterior

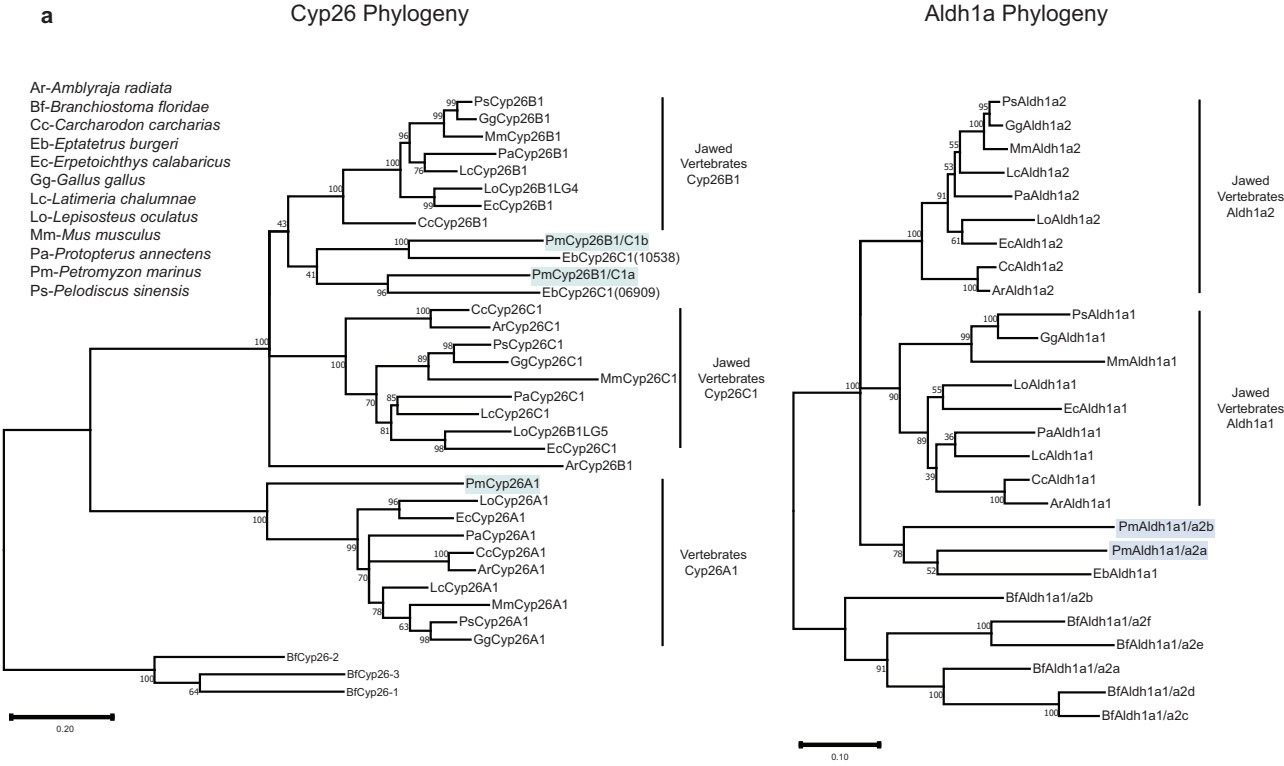

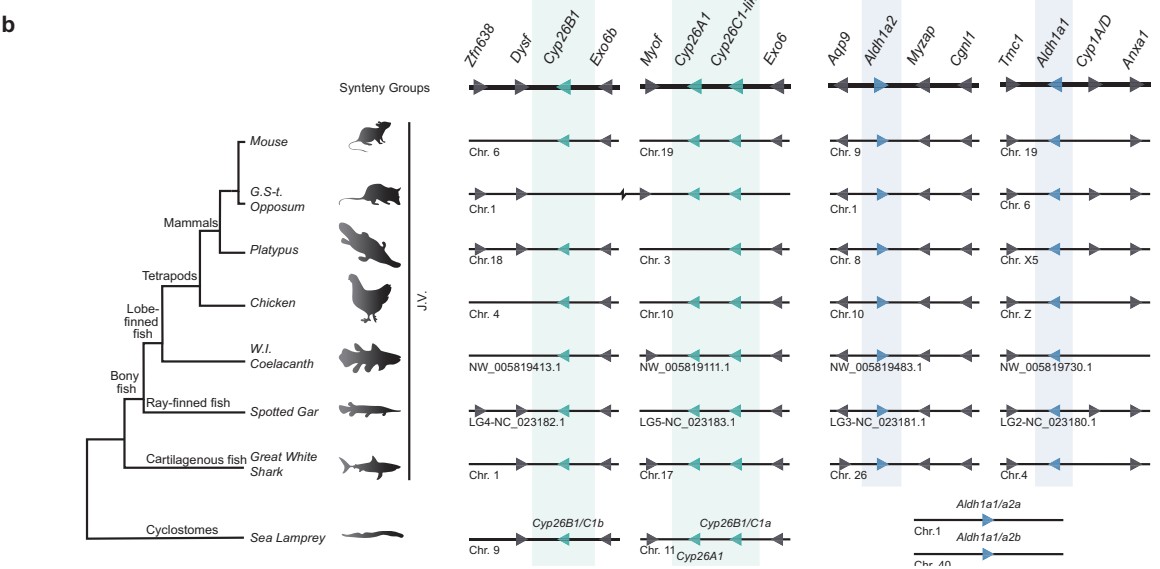

**Fig. 2 | Phylogenetic and synteny analyses of the vertebrate *Cyp26* and *Aldh1a* complements. a** Phylogenetic analysis of the sea lamprey *Cyp26* and *Aldh1a* complements with Amphioxus *Cyp26* and *Aldh1a* homologs as outgroups respectively. Jawed Vertebrate (J.V.) *Cyp26B1/Cyp26C1*, J.V. *Aldh1a1/Aldh1a2*, and Vertebrate (V.) *Cyp26A1* clades are indicated with black lines. Trees were generated by Maximum Likelihood using the WAG model with 500 iterations for bootstrap testing, and the resulting supporting value for each node is shown as a percentage. A scale bar for the evolutionary distance is indicated below each tree. Species name abbreviations are indicated on the top left of the panel; **b** Schematic illustrating the

synteny analysis of the *Aldh1a* and *Cyp26* gene complements conducted for major jawed vertebrate lineages (J.V.) and the sea lamprey. The corresponding direction of transcription is represented with an arrow. Genes located in proximity to the *Cyp26* and *Aldh1a* genes and used for this analysis as well as their direction of transcription are represented by gray arrows. Synteny groups with gene names are represented at the top of the panel (in bold). Chromosome numbers, scaffold or linkage groups (LG) names are indicated below the loci. Sea lamprey *Cyp26* and *Aldh1a* complements are color-coded in teal and blue respectively.

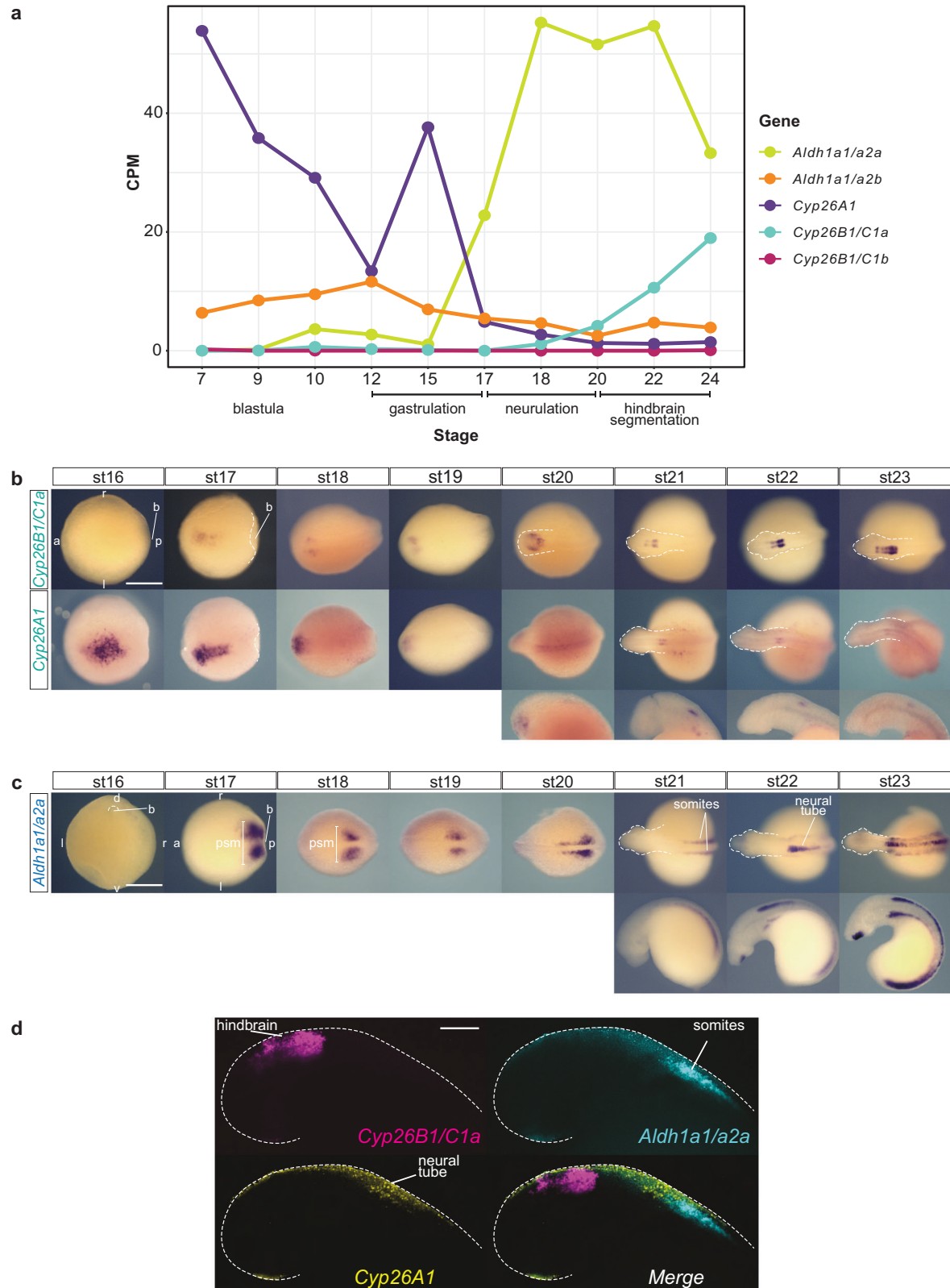

part of the trunk. During later stages (st22-st23), *Aldh1a1/a2a* expression persists in the somitic mesoderm and PSM, but also begins to appear in the dorsal spinal cord and in mouth tissue (Fig. 3c). The expression in somite and PSM domains is highly similar to the expression of *Aldh1a2* in zebrafish and mouse embryos[55–57]

where it serves as a primary source of RA synthesis required for regulation of patterning in the developing hindbrain[14,36,53,58,59]. *Aldh1a1/a2a* expression in lamprey paraxial mesoderm opens the possibility that it may play a similar role in generating RA important for regulating hindbrain segmentation.

**Fig. 3 | RNA seq profile and gene expression analysis of sea lamprey *Cyp26A1*, *Cyp26B1/C1a* and *Aldh1a1/a2a* in wild type embryos. a** RNAseq profile of sea lamprey *Cyp26A1*, *Cyp26B1/C1a* and *Aldh1a1/a2a* at blastula (-st9), gastrulation (st12-st15), neurulation (st15/16) and hindbrain segmentation (st20-24); **b** mRNA expression of sea lamprey *Cyp26A1*, *Cyp26B1/C1a* revealed by cISH. The orientation of the embryo, anterior (a), posterior (p), left (l) and right (r) is indicated on the image. *Cyp26A1* expression from st20 to st23 is shown with both dorsal view of the whole embryo (top) and lateral (bottom) view of the embryo's head. The scale bar corresponds to approximately 500 μm; **c** mRNA expression of sea lamprey *Aldh1a1/*

*a2a* revealed by cISH, expression at st21 to st23 is shown with both dorsal (top) and lateral (bottom) view of the whole embryo. Various embryological structures are indicated such as the blastopore (b) and the presomitic mesoderm (psm); The scale bar corresponds to approximately 500 μm; **d** HCR image showing the relative position of *Cyp26B1/C1a* (magenta), *Cyp26A1* (yellow) and *Aldh1a1/a2a* (cyan) expression domains at stage 22. The scale bar corresponds to approximately 125 μm. In panels of (**b**–**d**) ≥5 wild type embryos were assayed for each stage. stage (st).

## Expression of *Cyp26s* and *Aldh1a* during hindbrain segmentation

Given the dynamic expression of *Cyp26A1*, *Cyp26B1/C1a* and *Aldh1a1/a2a* in mesodermal and neural territories, we next investigated how their spatial and temporal patterns of expression relate to each other and whether any of these are correlated with specific rhombomeres. We optimized a protocol for Hybridization Chain Reaction fluorescent in situ hybridization (HCRv3-FISH)[60] (Supplementary Fig. 4), to visualize and directly compare the expression of these genes.

During early hindbrain segmentation (st21), *Cyp26A1* (yellow) is restricted to the most anterior part of the spinal cord, in a domain abutting the somitic expression of *Aldh1a1/a2a* (cyan) (Fig. 3d). This spatial relationship is reminiscent of that seen for jawed vertebrate models, where RA is synthesized in the somites by Aldh1a2 and degraded anteriorly in the hindbrain by *Cyp26B1* and *Cyp26C1*[36,53–57]. Thus, the expression of lamprey *Cyp26A1* in a domain adjacent to *Aldh1a1/a2a*, positions it for a potential role in modulating RA levels as it spreads more anteriorly in the neural tube.

*Cyp26B1/C1a* (magenta) is strongly expressed in the hindbrain, anterior to *Cyp26A1* (Fig. 3d). In order to determine whether the stripes of *Cyp26B1/C1a* expression align with developing rhombomeres, we performed HCRv3-FISH assays using probes for *Cyp26B1/C1a* and *krox20*, a marker of r3 and r5, which is initially expressed in r3 and then in r3/r5 in lamprey and other vertebrates (Fig. 4a)[34,61–63]. The initial neural domain of *Cyp26B1/C1a* expression (magenta) (st20) directly overlaps with *krox20* (cyan) in r3 and by st21 both display segmentally-restricted expression in r3 and r5. By st22, when segmentation is nearing completion, *Cyp26B1/C1a* expression covers a region from r1 to r6, with higher levels in r3, r5 and r6 compared to r1-2 and r4 (Fig. 4a, b).

The expression dynamics of lamprey *Cyp26* genes appear to support conservation of similar phases to those seen in gnathostomes, when considered in relation to morphological events and the expression of other patterning genes. Anterior *Cyp26A1* expression between gastrulation and neurulation (st16-20) coincides with the onset of expression of genes such as *HoxPG1*, *vHnf1*, and *kreisler*, which in gnathostomes are involved in formation of molecular subdivisions that prefigure rhombomeres[14,53]. Later segmental expression of *Cyp26B1/C1a* at st21-23 in lamprey coincides with the period in which rhombomeres become visible morphologically, and when *krox20*, *kreisler* and *Hox* genes are segmentally expressed[34,42]. Thus, it appears that the successive stages in the formation of subdivisions and sharpening or refining these boundaries seen in gnathostomes such as zebrafish, are also reflected in the expression of the lamprey *Cyp26* genes during hindbrain development.

Taken together, these patterns of spatio-temporal gene expression in lamprey embryos are consistent with potential roles for the *Aldh1a1/a2a*, *Cyp26A1* and *Cyp26B1/C1a* genes in regulating levels of RA that impact hindbrain segmentation.

## Modulation of RA levels impacts hindbrain segmentation

To investigate whether RA signaling is functionally involved in regulating hindbrain segmentation in lamprey, we utilized pharmacological inhibitors of the Aldh1a2 and Cyp26 enzymes to perturb endogenous levels of RA. This approach has previously been used to

characterize roles for RA signaling during hindbrain segmentation in zebrafish[53,64]. Based on the timing of expression of *Aldh1a1/a2a* and *Cyp26* genes (Fig. 3b–d), we used the following experimental design: batches of early gastrula embryos (st13) were grown until the end of hindbrain segmentation (st23), in media containing: i) 10 μM Talarazole, an inhibitor of Cyp26 enzymes that increases levels of RA; ii) 50 μM DEAB, which inhibits Aldh1a2 enzymes and reduces levels of RA synthesis; or iii) 0.1% DMSO, as a control (Fig. 5a). Given that rhombomeres are faint and relatively difficult to visualize in lamprey, we chose to use molecular markers such as *krox20* and *kreisler* and several *Hox* genes to mark rhombomeres in our study. These markers have been used in previous studies[34,39,42,63] and provide a readout of the molecular regionalization that demarcates neuroepithelial segments. To assess the effect of these treatments on hindbrain segmentation, we performed cISH on the drug-treated embryos using probes for key genes involved in hindbrain patterning: a) *otx* and *wnt1* mark subdivision of the forebrain and midbrain[26]; b) *krox20* and *kreisler* are segmental subdivision genes that mark the r3/r5 and r5/r6 territories, respectively[34,42]; and c) *hoxβ1*, *hoxa2*, *hoxa3*, *hoxζ4* are segmental identity genes each marking different groups of rhombomeres (Fig. 5a)[42].

In DEAB-treated embryos, the *otx* and *wnt1* expression domains define clear forebrain- and midbrain territories, that appear unaltered compared to control embryos (Fig. 5b, d). In contrast, expression of the segmental subdivision genes, *krox20* and *kreisler*, is no longer detected in the neural tube (Fig. 5d). Furthermore, none of the 4 selected *hox* genes are expressed in the hindbrain or the spinal cord of DEAB-treated embryos, with the exception of *hoxa2*, which maintains a faint anterior domain of expression in the hindbrain that may represent an r2-like region (Fig. 5d). These data (summarized in Fig. 6b) imply that proper levels of RA are required for activating and/or maintaining regulatory modules of the hindbrain GRN necessary for subdividing the hindbrain into segments and patterning their distinct identities via the *Hox* genes, at least posterior to r2.

In Talarazole-treated embryos, *otx* and *wnt1* exhibit major changes in their expression domains (Fig. 5b, c). *otx* is restricted to a small faint domain of expression in the anterior part of the head, while *wnt1* expression is reduced and shifted more anteriorly, such that its anterior boundary now roughly corresponds to the most anterior part of the neural tube. This indicates that elevated levels of RA cause a reduction in size of the forebrain and midbrain territories which are shifted anteriorly to a small region located at the tip of the head.

There are also significant changes in the expression of genes regulating segment formation (Fig. 5c). *krox20* is expressed in two small stripes, corresponding to r3- and r5-like territories. The posterior r5-like stripe of expression of both *krox20* and *kreisler* is very faint and difficult to detect, implying a major reduction in the size of the r5-like territory. These data imply that RA may regulate the genetic programs setting up r3 and r5 in different ways, which is in line with evidence in jawed vertebrates indicating that different mechanisms are involved in the regulation of *Krox20* in r3 and r5[62,65–68]. There are also dramatic perturbations in the normal nested and segmental patterns of *hox* expression in the hindbrain. In control embryos, *hox* genes have defined anterior limits of expression in specific rhombomeres (Fig. 5a) but upon exposure to Talarazole, the expression of all 4 *hox* genes is

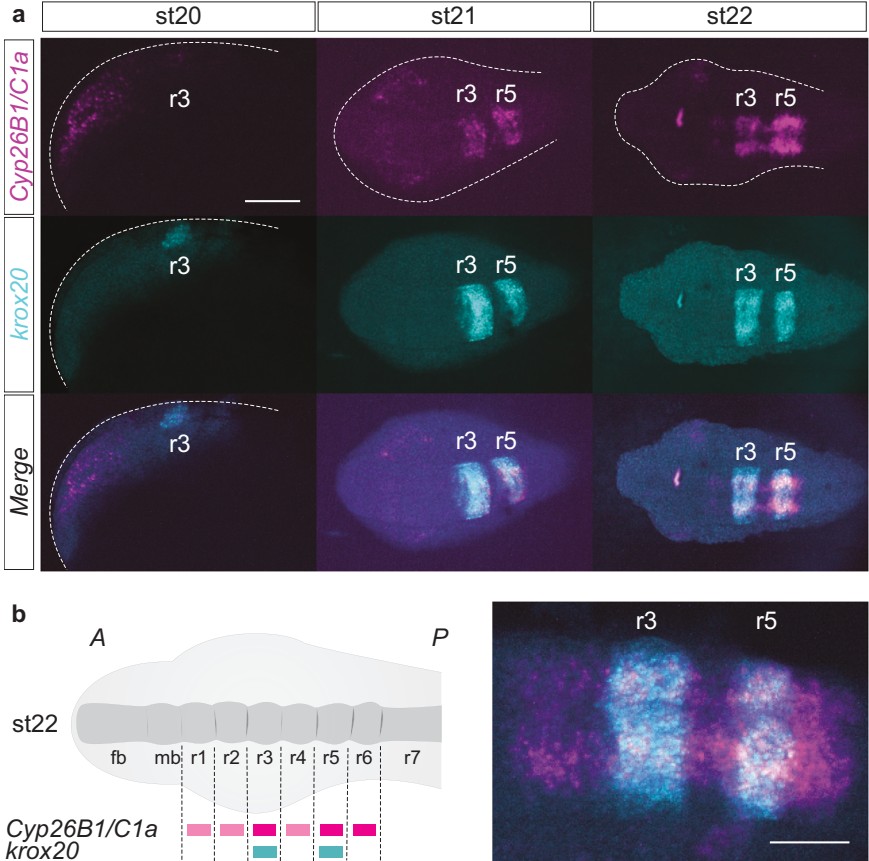

**Fig. 4 | *Cyp26B1/C1a* is directly coupled to hindbrain segmentation in the sea lamprey. a** HCR images showing the individual expression patterns of *Cyp26B1/C1a* (magenta) and *krox20* (cyan) as well as their merged expression during the process of hindbrain segmentation (st20 to st22). Rhombomeres (r) 3 and 5 are numbered. st20 is a lateral view of the developing head while st21 and st22 are dorsal views of the developing head. The scale bar corresponds to approximately 170 μm; **b** Cartoon summarizing the segmental expression of *Cyp26B1/C1a* at stage 22; zoomed HCR image of *Cyp26B1/C1a* (magenta) and *krox20* (cyan) expression at stage 22. The scale bar corresponds to approximately 100 μm. stage (st).

dramatically upregulated, shifted anteriorly, and lack segmental stripes seen in control embryos (Fig. 5b, c). This is consistent with the malformation of segments observed by loss of expression of segmental subdivision genes. Hence, in lamprey the formation and patterning of the developing fore-, mid- and hind- brain are strongly affected by the inhibition of Cyp26 activity, as summarized in Fig. 6b. Together, these experiments with drug treatments illustrate the need to maintain the proper balance between RA synthesis and degradation for regulating hindbrain segmentation.

**Changes in segmental patterning in drug-treated embryos**
To gain a better understanding of the specific changes in segmental gene expression at the level of r3, r4 and r5 observed in Talarazole-treated embryos we used the adapted lamprey HCRv3 protocol with markers for these rhombomeres: *hoxβ1* (r4, magenta), *krox20* (r3/r5, cyan) and *kreisler* (r5, yellow) (Fig. 6a, Supplementary Fig. 4). We observe a faint dispersed expression domain of *kreisler* that overlaps with a weak posterior patch of *krox20* expression, indicating that this domain corresponds to a small residual r5-like territory (Fig. 6a). Therefore, the most anterior domain of *krox20* expression corresponds to an r3-like territory and is less severely affected by changes in levels of RA than the r5-like domain. *hoxβ1* expression in the spinal cord is shifted anteriorly, but we do not detect a clear r4-like stripe adjacent to the r5-like domain of *krox20* (Fig. 6a, Supplementary Fig. 4), suggesting the putative r4 territory may have adopted an indeterminate fate. These changes are summarized in Fig. 6b and

highlight the impact that elevated levels of RA have in altering programs that regulate the formation of segments and segmental identity.

To further investigate the fate of r5, we combined the use of a transgenic GFP-reporter assay and pharmacological treatments. An enhancer from the zebrafish *hoxb3a* gene (*Drhoxb3a*) contains binding sites for Kreisler and Krox20 TFs and mediates GFP-reporter activity in r5 of transgenic lamprey embryos (DMSO panel in Fig. 6c)[34]. In Talarazole-treated transgenic embryos, r5 reporter activity is much fainter and shifted anteriorly (Fig. 6c), consistent with the anterior shifts previously seen for *krox20* and *kreisler* expression (Fig. 5c). Conversely, DEAB-treated transgenic embryos show no reporter activity in r5 (Fig. 6c), consistent with the loss of expression of *kreisler* and *krox20* in DEAB-treated embryos (Fig. 5d). This regulatory assay further suggests that a small r5-like domain persists in Talarazole-treated embryos. It also reveals that the impact of RA perturbation on the early steps of segment formation has important consequences on the activation of the molecular and cellular programs that work downstream in the progressive process of segmentation.

Collectively, these pharmacological perturbations highlight the importance of maintaining appropriate levels of RA for regulation of the cellular and molecular processes that govern hindbrain segmentation in lamprey. The experiments suggest that RA plays important roles in multiple aspects of the GRN for hindbrain segmentation in the sea lamprey: 1) in delimiting the future hindbrain territory; 2) in setting up the formation of segments (segmental subdivision via *kreisler* and *krox20* expression); and 3) in defining the molecular identity of each of

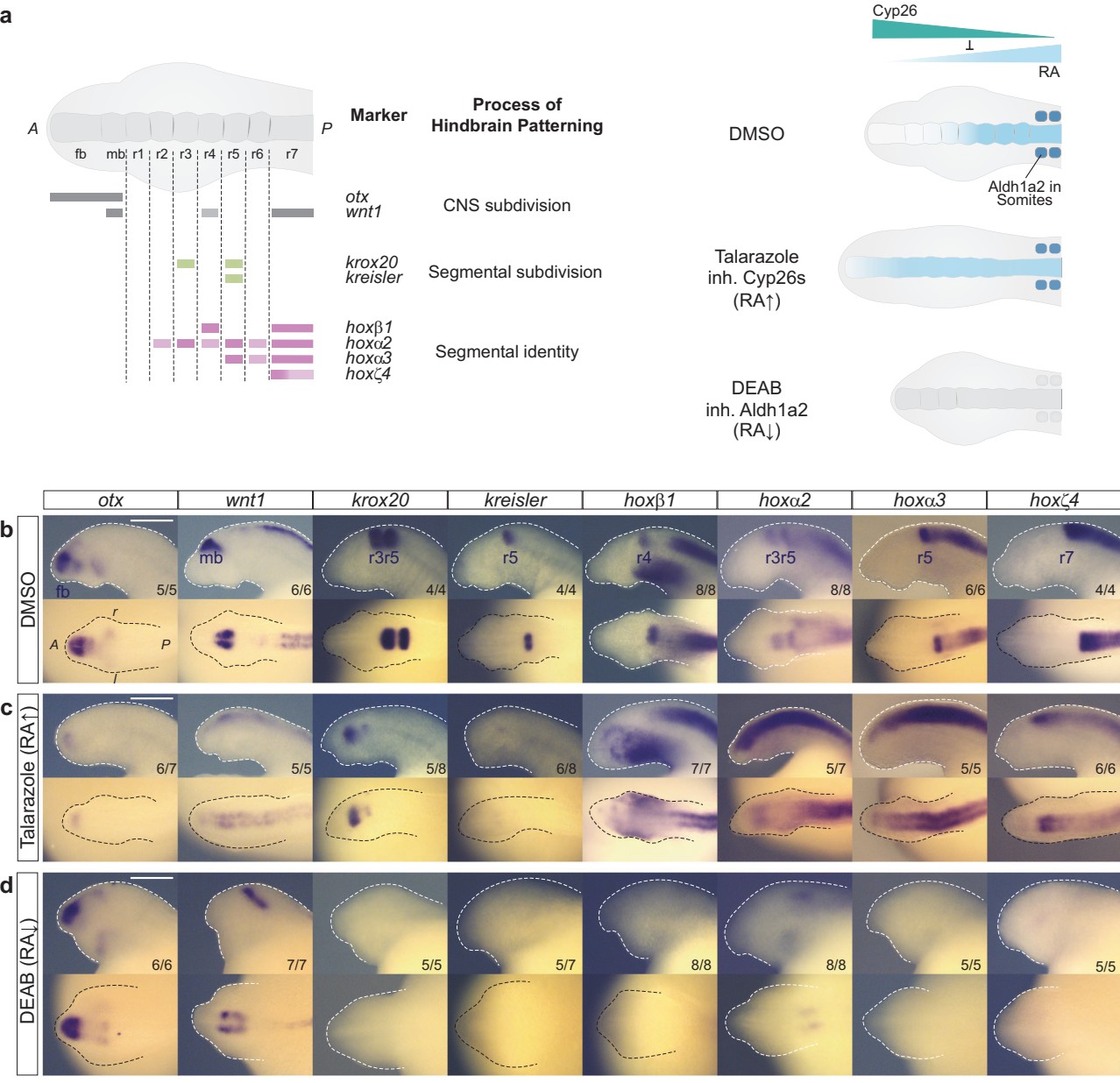

**Fig. 5 | Inhibition of synthesizing and degrading enzymes alters segmental patterning in the sea lamprey. a** Cartoons illustrating the different gene markers used to reveal various aspects of hindbrain patterning, as well as the effect of Talarazole in inhibiting the activity of the Cyp26 enzymes leading to increased levels of RA (less degradation) while the use of DEAB inhibits the activity of Aldh1a2, leading to reduced levels of RA synthesis. cISH of key patterning hindbrain markers in embryos treated with DMSO (**b**), Talarazole (**c**) and DEAB (**d**). *Otx* and *wnt* define the hindbrain territory within the developing CNS (CNS subdivision), *krox20* and *kreisler* participate in the early process of making segments as part of the segmental subdivision module of the hindbrain GRN and *hoxβ1*, *hoxα2*, *hoxα3*, *hoxζ4* contribute to the segmental identity aspect of the hindbrain GRN. For each drug condition, lateral and dorsal views of stage 23 embryos are shown on top and bottom rows respectively (only heads are shown). For each gene, the most representative ISH phenotype is shown, and numbers of experimental replicates are indicated. Important features are indicated such as the forebrain (fb), midbrain (mb) and rhombomeres (r) of the hindbrain. The scale bars **b**–**d** correspond to approximately 500 μm.

these segments via *hox* expression. The changes we observed are analogous to those seen in jawed vertebrate embryos treated with these drugs[53,54,64,69], implying that there are similar regulatory inputs from RA into the hindbrain GRN in all vertebrates.

### *Cyp26s* are required for hindbrain segmentation in lamprey

Because pharmacological inhibitors may affect multiple proteins and have off-target or indirect effects, it is important to demonstrate that the changes in hindbrain patterning we observed after treatment with DEAB or Talarazole are directly linked to the *Cyp26A1, Cyp26B1/C1a* and *Aldh1a1/a2a* genes encoding these enzymes. Hence, we conducted

functional analyses of these genes in sea lamprey by implementing a CRISPR-mediated mutagenesis approach using guide RNAs (gRNAs) designed to target regions that encode key catalytic sites (Fig. 7a)[47,48]. We obtained robust and reproducible mutant phenotypes and sequenced the targeted regions of the genes in affected embryos to confirm their genotypes (Supplementary Figs. 5 and 6).

Following CRISPR-mediated mutagenesis of the *Cyp26A1* and *Cyp26B1/C1a* genes (Fig. 7b; Supplementary Fig. 5), we performed cISH analysis on CRISPR (Cr) embryos at st23 using marker genes to investigate the formation of segments (*krox20* and *kreisler*) and segmental identity (*hoxβ1* and *hoxζ4*) (Fig. 7c). When individual genes,

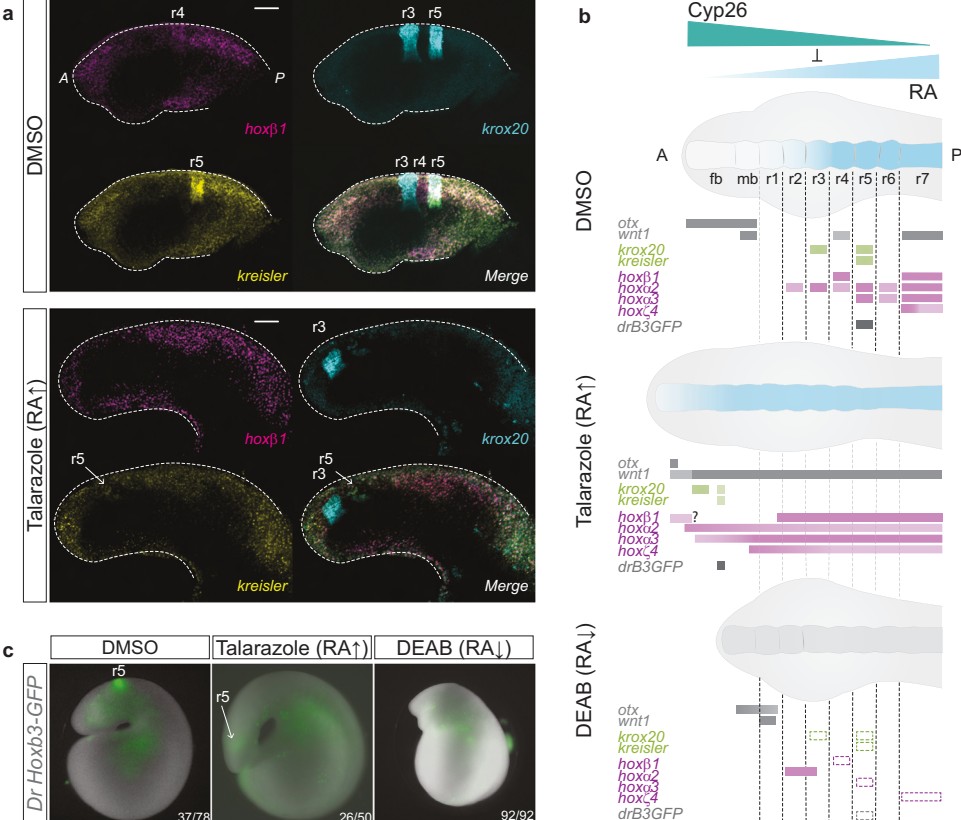

**Fig. 6 | Consequences of inhibiting RA synthesis and degradation on the hindbrain GRN for segmentation. a** HCR-FISH images showing the individual expression patterns of *hoxβ1* (magenta), *krox20* (cyan) and *kreisler* (yellow) as well as their merged expression in DMSO and Talarazole treated stage 23 embryos. Lateral views of the head are shown and rhombomeres (r) are indicated; ≥5 embryos were assayed for each treatment. The scale bar corresponds to approximately 115 μm; **b** Cartoon summarizing the expression of hindbrain markers based on cISH, HCR-FISH and GFP-reporter analysis in DMSO, Talarazole and DEAB treated embryos. Rectangles with dashed lines are used in DEAB to represent where

the expression of the gene would be expected in WT embryos. fb, forebrain; mb, midbrain; rhombomere (r); **c** Image of live *Drhoxb3a*-GFP stage 23 sea lamprey reporter embryos in DMSO, Talarazole and DEAB treated embryos. *Drhoxb3a*-GFP reports activity of the enhancer in rhombomere 5 (r5). Images shown represent the most common phenotype of GFP reporter expression (~50% for DMSO, 50% for Talarazole and 100% for DEAB embryos). Corresponding ratio of positive embryos are indicated (number of embryos showing the phenotype/number of GFP positive embryos).

CrCyp26A1 or CrCyp26B1/C1a, were targeted all of these markers are shifted anteriorly. The relative levels of *krox20* and *kreisler* appear similar to that observed in control embryos, but there is reduced expression of *hoxβ1* and *hoxζ4* and loss of a clear r4 stripe of *hoxβ1* expression (Fig. 7c). The molecular and morphological phenotypes of the CrCyp26A1 and CrCyp26B1/C1a embryos are very similar, suggesting that these two *Cyp26* genes may functionally compensate for each other when one of them is disrupted. Hence, we targeted both *Cyp26A1* and *Cyp26B1/C1a* (CrCyp26s) in individual embryos, which generated a more severe phenotype with a dramatic loss of *krox20, kreisler* and *hoxβ1* expression, and an anterior shift of *hoxζ4* (Fig. 7c). This shows that both *Cyp26A1* and *Cyp26B1/C1a* are necessary for regulating endogenous levels of RA in the developing hindbrain, as in the absence of these two genes expression of segmental markers is completely lost (Fig. 7c CrCyp26s). *Cyp26* genes also play partially redundant roles in mouse and zebrafish hindbrain patterning[36,53].

The molecular phenotype observed in CrCyp26s embryos is similar to but more severe than the one observed in Talarazole-treated embryos, where the Cyp26 activity is inhibited (Fig. 7d). This difference may be explained by the timing of the drug treatment and/or the degree of impact on enzymatic activity, as compared to the CRISPR embryos, which would be expected to have little or no Cyp26 activity. The extensive molecular phenotypes observed in the CrCyp26s embryos clearly show that *Cyp26* genes play key roles in regulating

levels of RA that have important inputs into the GRN for hindbrain segmentation in lamprey.

**Aldh1a1/a2a contributes to hindbrain segmentation in lamprey**
We targeted *Aldh1a1/a2a* for functional perturbation by designing gRNAs to delete regions around the Cysteine (Cys) and Glutamic acid (Glu) residues in the conserved catalytic sites responsible for enzymatic activity (Fig. 8a, b; Supplementary Figs. 2 and 6)[48]. Cr*Aldh1a1/a2a* embryos exhibit severely truncated heads that morphologically appear very similar to DEAB-treated embryos (Fig. 8c). We detected a posterior shift in the expression domains of *krox20* and *kreisler* and elongation of caudal hindbrain segments. There is also a complete loss of *hoxζ4* expression and a posterior shift and elongation of *hoxβ1* expression domain in r4. Thus, CRISPR-mediated disruption of *Aldh1a1/a2a* alters segmental patterning, particularly in the r5-r7 region. This demonstrates roles for RA via *Aldh1a1/a2a* in regulating *hoxζ4* and *hoxβ1*, as well as providing inputs into the regulation of *kreisler* and *krox20* in r5.

The genetic disruption of *Aldh1a1/a2a* induces a relatively milder mis-patterning of the hindbrain compared with DEAB treatment (Figs. 5c and 6b) and the Cr*Aldh1a1/a2a* phenotype is very similar to that observed in embryos treated with a lower concentration of DEAB (10 μM) (Supplementary Fig. 7). This is analogous to findings in zebrafish, where the *nls/raldh2 (Aldh1a2)* mutant phenotype is not as

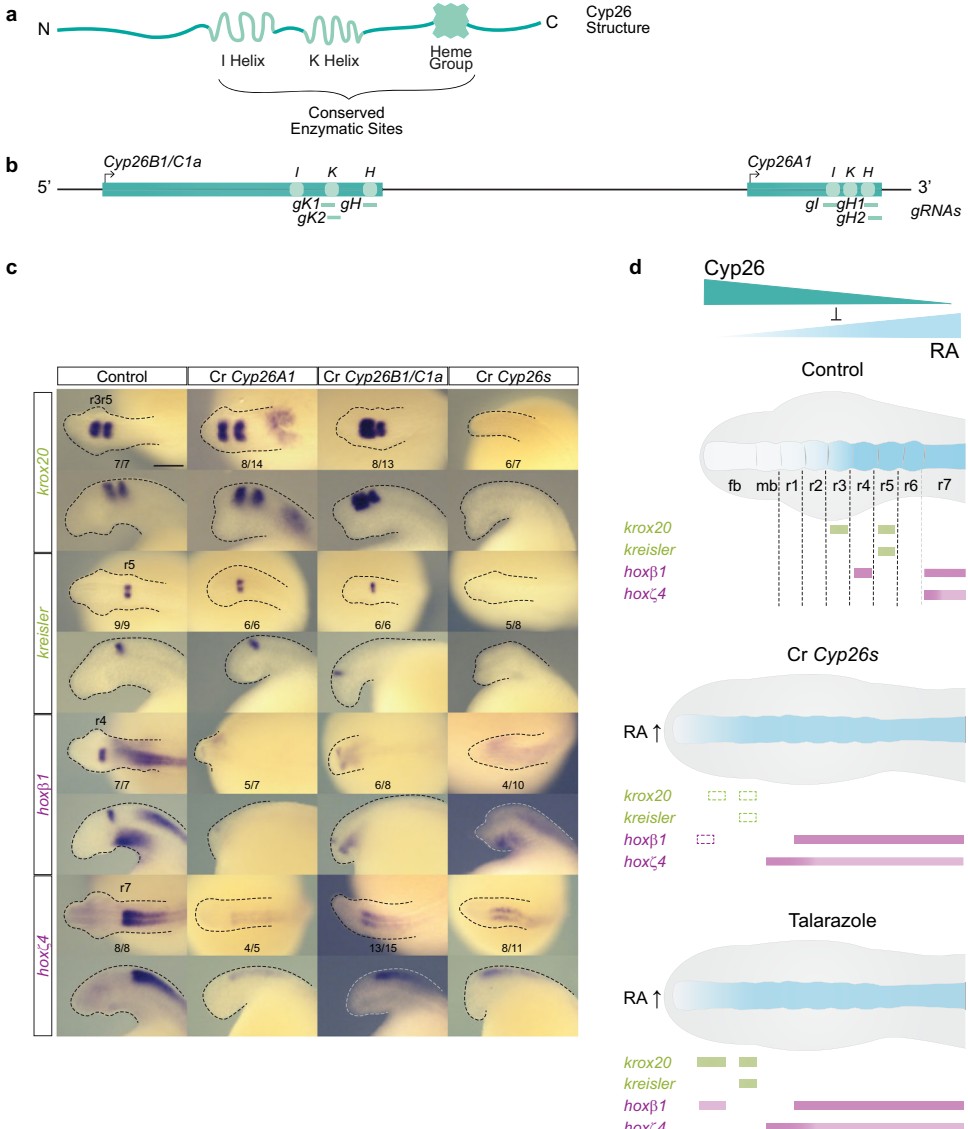

**Fig. 7 | Both *Cyp26A1* and *Cyp26B1/C1a* are required for hindbrain segmentation in the sea lamprey. a** Schematic representation of the general structure of a Cyp26 protein with its highly conserved enzymatic sites, I helix, K Helix and Heme; **b** Schematic representation of the sea lamprey *Cyp26* locus which includes *Cyp26B1/C1a* and *Cyp26A1* genes, and the sequences targeted by CRISPR/Cas9 gRNA corresponding to the highly conserved enzymatic sites; **c** cISH of key patterning hindbrain markers in CRISPR treated embryos: *krox20* and *kreisler* are used as markers of segmentation (green) while *hoxβ1* and *hoxζ4* are used as markers of segmental identity (purple). For each gene, the most representative ISH phenotype is shown, and numbers of experimental replicate are indicated. Lateral and dorsal views of stage 23 embryos are shown on the top and bottom rows respectively (only heads are a shown). The scale bar corresponds to approximately 500 μm. Rhombomeres (r) are indicated; **d** Cartoon summarizing the expression of hindbrain markers in Cyp26 CRISPR (Cr *Cyp26s*) embryos. Rectangles with dashed lines are used in the Cr *Cyp26s* panel to represent where the expression of the gene would be expected in WT embryos. fb forebrain, mb midbrain, rhombomere (r); Cartoon summarizing the molecular phenotypes of gene markers of hindbrain segmentation obtained by cISH in Talarazole treated embryos (in which the Cyp26s are inactivated), fb forebrain, mb midbrain.

severe as those observed in DEAB-treated embryos[55,64]. This suggests that zebrafish Aldh1a2 and sea lamprey Aldh1a1/a2a may not be the sole source of RA synthesis, contrasting with mouse where Aldh1a2 appears to be the main source of RA synthesis[23]. Collectively these results imply the presence of additional Aldh inputs into RA synthesis during hindbrain segmentation that may be inhibited by DEAB acting as a pan-Aldh inhibitor[70].

Using gene editing approaches, we have generated independent evidence on functional roles of the *Cyp26A1*, *Cyp26B1/C1a* and *Aldh1a1/a2a* genes that are consistent with conclusions from our analyses using pharmacological treatments. Collectively these data demonstrate that RA signaling is coupled to the GRN for hindbrain segmentation in sea lamprey.

## Cyp26s and Aldh1a1/a2a respond to changes in levels of RA

In jawed vertebrates, some genes involved in the synthesis and degradation of RA are directly regulated by RA. In zebrafish, mouse and human models, *Cyp26A1* is directly regulated by RA via a conserved set of *RAREs (R1, R2 and R3)* located in its promoter region[71–74]. Studies in zebrafish also point to a role for RA in repressing the expression of *Aldh1a2*, as applying exogenous RA leads to reduced activity of its promoter region and there is an upregulation of *Aldh1a2* transcription in the *nls/raldh2* mutant[55,75]. These data illustrate that in jawed vertebrates, feedback mechanisms play an important role in regulating levels of RA by controlling its synthesis and degradation. Hence, we investigated whether lamprey *Cyp26A1*, *Cyp26B1/C1a* and *Aldh1a1/a2a* are responsive to changes in levels of RA during hindbrain

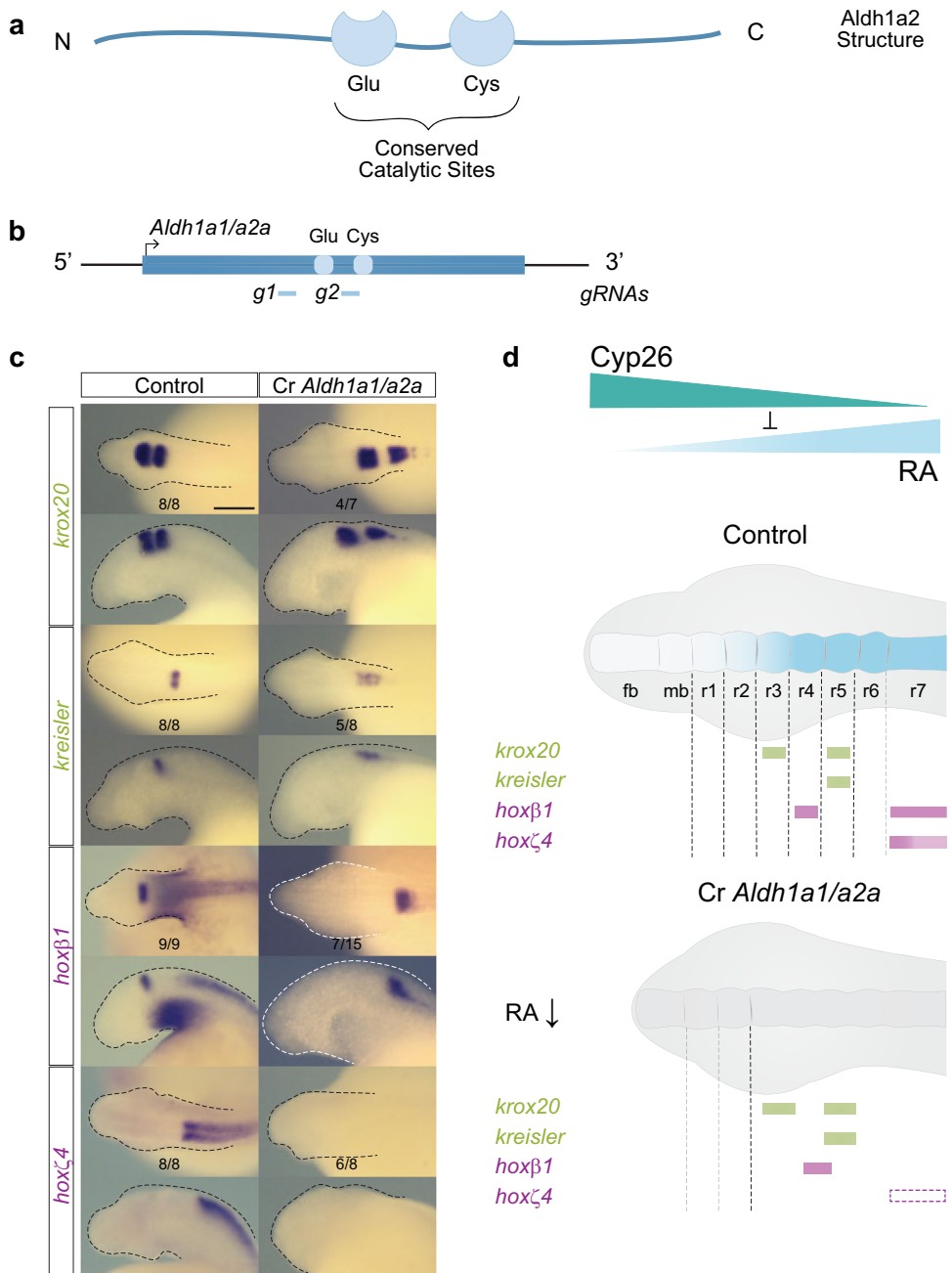

**Fig. 8 | *Aldh1a1/a2a* influences hindbrain segmentation in the sea lamprey.**
**a** Schematic representation of the general structure of Aldh1a2 with its highly conserved catalytic sites, Glutamate (Glu) and Cysteine (Cys); **b** Schematic representation of the sea lamprey *Aldh1a1/a2a* gene, and the sequences targeted by CRISPR/Cas9 gRNA corresponding to the conserved Glu and Cys sites; **c** cISH of key patterning hindbrain markers in CRISPR treated embryos: *krox20* and *kreisler* are used as markers of segmentation (green) while *hoxβ1* and *hoxζ4* are used as markers of segmental identity (purple). For each gene, the most representative ISH phenotype is shown, and numbers of experimental replicate are indicated. Lateral and dorsal views of stage 23 embryos are shown on the top and bottom rows respectively (only heads are a shown). The scale bar corresponds to approximately 500 μm. Rhombomeres (r) are indicated; **d** Cartoon summarizing the expression of hindbrain markers in *Aldh1a1/a2a* CRISPR (*Cr Aldh1a1/a2a*) embryos. Rectangles with dashed lines are used in the *Cr Aldh1a1/a2a* panel to represent where the expression of the gene would be expected in WT embryos. fb forebrain, mb midbrain, rhombomere (r).

segmentation. Increasing the levels of RA by exposure to Talarazole leads to reduced expression of *Aldh1a1/a2a*, while reduced levels of RA synthesis mediated by DEAB lead to increased expression of *Aldh1a1/a2a* (Fig. 9a). The expression of *Cyp26A1* is lost in DEAB-treated embryos and is strongly increased in Talarazole-treated embryos, suggesting that RA has an important and potentially direct role in initiating and maintaining its expression. *Cyp26B1/C1a* expression is also affected by the drug treatments, as in both cases the domain of

expression appears smaller and weaker than in control embryos (Fig. 9b). These changes in expression, summarized in Fig. 9c, are consistent with the predicted differences in segmental organization induced by the drug treatments (Figs. 5 and 6) and hence are likely to be indirect. Our data indicate that the negative regulation of *Aldh1a1/a2a* and the positive regulation of *Cyp26A1* by RA are important aspects of feedback regulation of RA synthesis and degradation during hindbrain segmentation in all vertebrates (Fig. 9d).

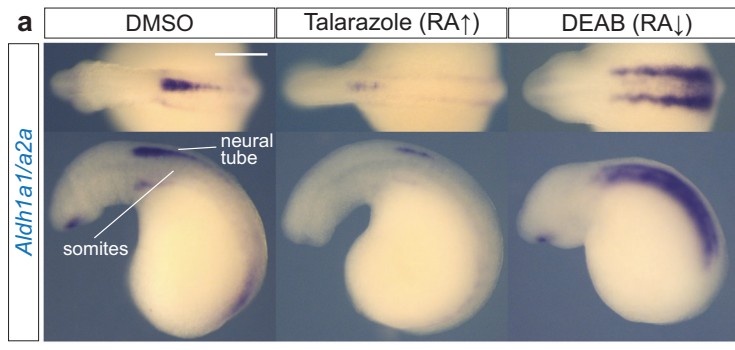

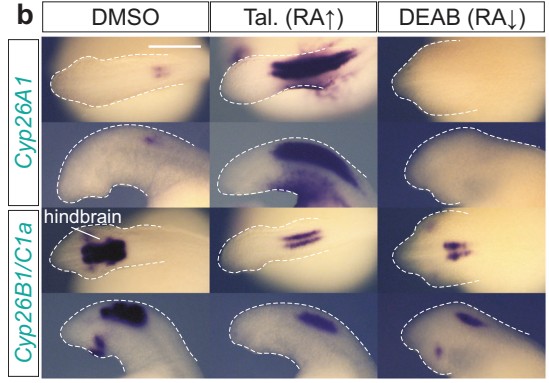

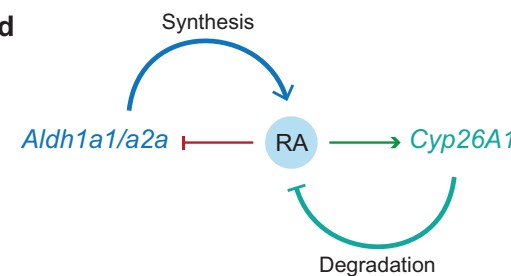

**Fig. 9 | Sea lamprey *Aldh1a1/a2a*, *Cyp26A1*, *Cyp26B1/C1a* respond to drug-induced changes in the levels of RA.** cISH of (**a**) *Aldh1a1/a2a* and (**b**) *Cyp26A1* and *Cyp26B1C1a* in DMSO, Talarazole and DEAB treated embryos. For each drug condition, lateral and dorsal views of stage 23 embryos are shown on the top and bottom rows respectively (only heads are a shown). For each gene, 4 embryos were used per treatment condition, and all showed the phenotype shown in the image; The scale bar corresponds to approximately 500 µm; **c** Cartoon summarizing the gene expression in each treatment condition. Rectangles with dashed lines are used in the DEAB cartoon to represent where the expression of the gene would be expected in WT embryos. fb forebrain, mb midbrain, rhombomere (r); **d** Cartoon illustrating mechanisms of self-regulation of the RA machinery where RA down-regulates the expression of *Aldh1a1/a2a*, while up regulating the expression of *Cyp26A1*, therefore exerting a control on its own synthesis and degradation.

## Discussion

We have investigated the origin and evolution of the coupling of RA to hindbrain segmentation in vertebrates, using sea lamprey as a model. We demonstrated that components of the RA signaling pathway mediating RA synthesis (*Aldh1as*) and degradation (*Cyp26*s) are expressed in a spatio-temporal manner consistent with roles in regulating hindbrain segmentation in lamprey. Notably, we found *Cyp26B1/C1a* expression aligns with specific rhombomeres. Using pharmacological treatments and CRISPR/Cas9 gene editing to perturb the activity of these components we showed that multiple aspects of hindbrain segmentation and patterning in lamprey require RA signaling. Furthermore, lamprey *Cyp26A1* and *Aldh1a1/a2a* genes themselves also appear to be regulated by feedback circuits involving RA signaling. Collectively, these results reveal that an ancestral RA/*Hox* regulatory circuit for axial patterning became coupled to the process of hindbrain segmentation prior to the split between jawed and jawless vertebrates

(Fig. 10). These findings raise interesting questions and avenues for further investigation.

Our analyses of the *Cyp26* genes in lamprey reveal similarities and differences in their coupling to hindbrain segmentation in jawed and jawless vertebrates (Supplementary Fig. 10). In both groups, *Cyp26A1* is expressed in the anterior neural plate during gastrulation, while *Cyp26B1*, *Cyp26C1* and *Cyp26B1/C1a* are expressed later in a dynamic and segmental manner across r2-r6 (Figs. 3 and 4)[14,36,53,54]. However, rhombomeric patterns vary significantly between vertebrate species (Supplementary Fig. 10). For example, lamprey *Cyp26B1/C1a* has lower levels of expression in r2/r4 compared to r3/r5/r6 (Fig. 4). In contrast, zebrafish *Cyp26b1* and *Cyp26c1* expression is higher in r2/r4/r6 than in r3/r5, where they are repressed by Krox20[53,76]. This dynamic segmental expression has been linked to important processes in hindbrain segmentation: a) the generation of shifting domains of RA that modulate the anterior limit of expression of genes that prefigure rhombomere

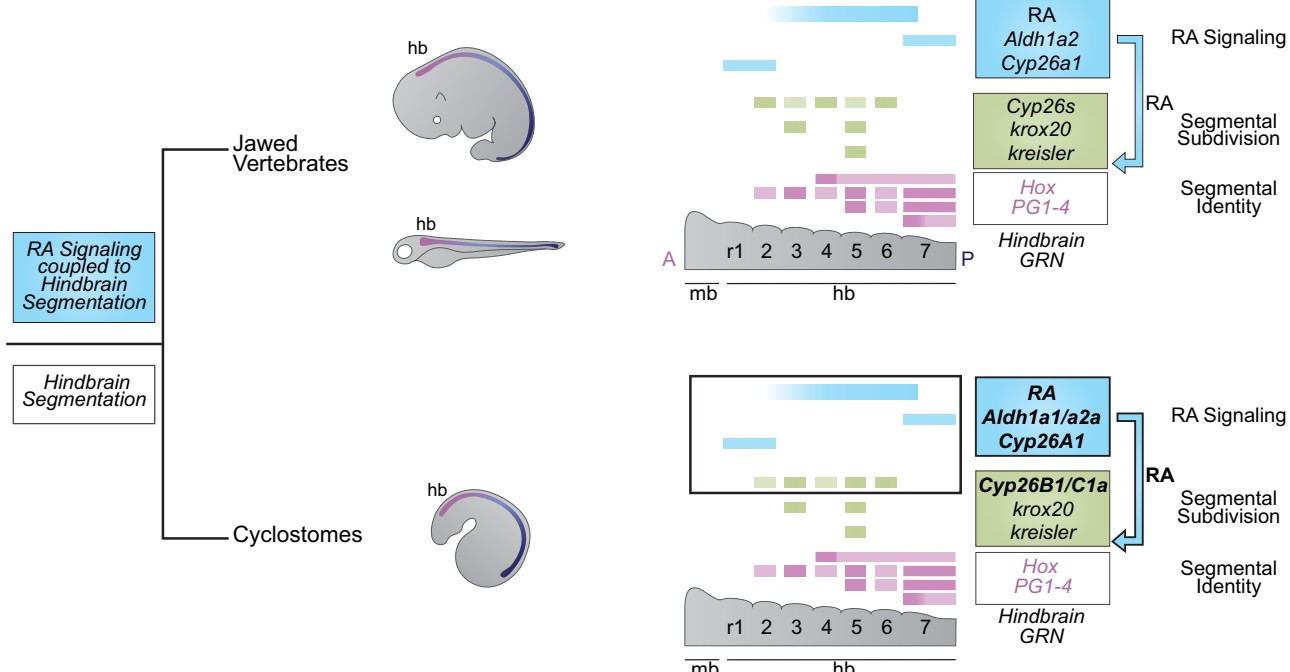

**Fig. 10 | The role of RA signaling in the GRN for vertebrate hindbrain segmentation is conserved to the base of vertebrates.** Using sea lamprey as a jawless vertebrate model, we show that the origin of the coupling of RA to the hindbrain GRN for segmentation occurred before the split between jawed vertebrates and cyclostomes as RA signaling is directly coupled to the hindbrain GRN for segmentation in the sea lamprey.

boundaries[53,54]; b) segmental variation in RA levels across odd versus even rhombomeres that influences formation of sharp segmental boundaries by community signaling[76,77]. Thus, variation in *Cyp26* rhombomeric domains between vertebrate species may reflect divergence in their precise roles in regulating hindbrain segmentation. It is unknown whether this variation in segmental expression of *Cyp26s* translates to different patterns of rhombomeric RA concentrations between species. Visualizing or measuring endogenous RA across the developing hindbrain in different models could be enlightening in this regard[78].

From an evolutionary perspective, the conserved deployment of A-P patterning genes, such as *Hox* genes, reflects an ancient and highly conserved GRN for axial patterning that was probably present in the bilaterian ancestor[2,3]. FGF and Wnt signaling are important in this axial patterning GRN[2,6,7], but the coupling of RA signaling to *Hox* genes and A-P patterning appears to have arisen later in the deuterostome lineage[16,26]. Indeed, invertebrate chordate studies revealed that ancestral chordates employed an RA/*Hox* regulatory circuit to pattern the neural territory[5,24,25,79] and here we show that the role of the ancestral RA/*Hox* regulatory hierarchy appears to have been elaborated in early vertebrates, becoming coupled to the process of hindbrain segmentation. This could have occurred via evolution of *cis*-regulatory inputs of RA into gene expression through novel or co-opted RAREs, as has been shown in the *Hox* clusters[17,25,26]. Supporting this idea, recent comparative genome-wide studies of open chromatin during development revealed more enhancers per gene in zebrafish than in amphioxus, implying an increase in regulatory complexity in vertebrates[41,80]. Some of these vertebrate-specific elements respond to multiple signaling pathways (RA, Wnt, and FGF), suggesting that an increase of interactivity between these pathways evolved in early vertebrates and may underlie increased tissue complexity in vertebrates[41]. Thus, it is likely that novel regulatory inputs from these signaling pathways into target genes played an important role in the evolution of the GRN for vertebrate hindbrain segmentation.

The role of RA signaling in vertebrate hindbrain segmentation could also be linked to the evolution of new expression domains of genes that shape and interpret RA concentrations along the A-P axis, such as *Cyp26s*, *rars*, and *Crabps*[26]. This idea is supported by the *Cyp26* genes, which exhibit shared and derived features between vertebrates and invertebrate chordates, such as amphioxus. For example, there is a conserved early role for Cyp26s acting as anterior sinks for shaping the RA gradient in chordates but their dynamic segmental expression during hindbrain development in vertebrates is not seen in amphioxus[81,82]. Since this segmental expression is linked to multiple aspects of rhombomere formation, it is important to understand how this arose in vertebrates as opposed to other chordates.

*Cyp26A1* is induced by RA in jawed vertebrates via multiple RAREs[71,73,83] and we have shown that lamprey *Cyp26A1* also requires RA for its expression (Fig. 9). Similar responses were found in amphioxus *Cyp26s*, and an ancient pan-deuterostome RARE has been implicated in this role[82], suggesting that the RA-dependent regulation of *Cyp26s* may be an ancestral feature of deuterostomes. Functional characterization of RAREs around the tandemly linked *Cyp26A1-Cyp26B1/C1a* genes in lamprey could help decipher ancestral versus vertebrate-specific RA-dependent regulatory mechanisms.

Feedback-regulation of the RA machinery via *Cyp26* genes may constitute an ancestral mechanism that was further elaborated in vertebrates. However, it is also possible that the rhombomere-specific expression of vertebrate *Cyp26* genes could be linked to vertebrate-specific inputs from segmentally expressed TFs, such as Krox20. Given the variation in rhombomeric *Cyp26* expression between vertebrates, such inputs could vary between models. For example, *Cyp26C1* expression in r4 is induced by RA in mouse embryos but not in zebrafish[53,54]. In lamprey, we found that *Cyp26B1/C1a* expression in the hindbrain does not depend upon RA, indicating that it requires other regulatory inputs to generate its expression in rhombomeres. Thus, it will be important to identify such inputs and elucidate how components of the RA machinery became coupled to hindbrain segmentation in lamprey.

It is notable that the vertebrate *Cyp26* gene complement has been shaped by both whole genome duplication and tandem gene duplication prior to the split between jawed and jawless vertebrates. This coincides with the evolution of the coupling of RA to hindbrain segmentation. Thus, the duplication of the *Cyp26* genes may have been important in the elaboration of their roles in the hindbrain GRN, perhaps by enabling functional regulatory divergence between the resulting paralogs. Furthermore, additional independent rounds of whole genome duplication have recently been inferred in each lineage after their divergence[84]. These may have provided even further scope for functional regulatory divergence of *Cyp26* genes in each lineage.

While our data supports a shared vertebrate ancestry of the influence of RA in the genetic program for hindbrain segmentation, we do not exclude the importance of derived features in hindbrain development between gnathostome and cyclostome lineages. Indeed, while lamprey hox expression is transiently connected with specific rhombomeres, some hox genes subsequently escape segmental restriction at later stages. For example, *hoxα3* is initially expressed up to the r4/r5 border at st22-23 but then spreads anteriorly in specific neuronal populations within r4[34,39,42]. This appears to be different from the situation in gnathostomes, where such escape of hox expression from segmentation has not been described. Thus, based on current evidence in lamprey, it appears that Hox genes are initially coupled with the RA-dependent formation and early patterning of rhombomeres but subsequently Hox expression domains are not all maintained in segmental register, which may alter later roles in branchiomotor neuron positioning.

This study reveals a key role for RA signaling in influencing hindbrain segmentation, starting during the early steps of gastrulation (st13) but what about later stages? We have found that lamprey embryos treated with 50 μM DEAB at later time points, early neurulation stages (st17), also display severe defects in hindbrain patterning (Supplementary Fig. 7). This suggests that as in gnathostomes, RA appears to have multiple and successive roles in regulating the hindbrain GRN in lamprey. It is also possible that RA might play additional roles in developmental processes happening after hindbrain segmentation has reached completion, including neural crest patterning and neurogenesis.

In summary, we have shown that the coupling of the RA/*Hox* regulatory hierarchy to the GRN for hindbrain segmentation is rooted to the base of vertebrates. This raises questions regarding the evolution of regulatory mechanisms linked to the GRN that enabled this coupling. It will also be important to decipher the nature and origin of RA-dependent regulatory mechanisms of components of the RA signaling machinery (e.g., *Cyp26* and *Aldh1a* genes). More generally, this work illustrates how using sea lamprey as a jawless vertebrate model can yield insights into the evolution of GRNs underlying novel vertebrate features.

## Methods
This research study complies with all relevant ethical regulations and was conducted in accordance with the recommendations in the Guide for the Care and Use of Laboratory Animals of the NIH and protocols approved by the Institutional Animal Care and Use Committees of the California Institute of Technology (lamprey, MEB Protocol: #IA23-1436).

### Sea lamprey genome and model organism used for comparative studies
Sea lamprey (*Petromyzon marinus*) *Aldh1a* and *Cyp26* gene sequences were retrieved from the JBrowse browser and NCBI database, using the most-recent version of the Sea lamprey germline genome (KPetMar1). We found two predicted *Aldh1a2-like* genes with the following GenBank accession numbers: *retinal dehydrogenase 2-like* LOC116940317 ('*Aldh1a1/a2a*') and *retinal dehydrogenase 2-like-*LOC116950758

('*Aldh1a1/a2b*'). We also found three predicted *Cyp26* genes with the following GenBank accession numbers: *Cyp26A1*-LOC116941264, *Cyp26B1-like*-LOC116941263 ('C*yp26B1/C1a*') and *Cyp26B1-like-*LOC116940441 ('*Cyp26B1/C1b*').

Phylogenetic analyses of the sea lamprey *Aldh1a* and *Cyp26* complements were conducted using key representatives of the vertebrate group. In light of the established evolutionary history of the *Aldh1a* gene family in vertebrates[49] and the key position of sea lamprey as part of the early-diverged group of jawless vertebrates, we included both vertebrate *Aldh1a1* and *Aldh1a2* gene families in the phylogenetic analysis. Within jawed vertebrates, great white shark (*Carcharodon carcharias*) and thorny skate (*Amblyraja radiata*) were used as a representative of the cartilaginous fish group (Chondrichthyes). Spotted gar (*Lepisosteus oculatus*) together with reedfish (*Erpetoichthys calabaricus*) rather than zebrafish were used as a representative of the ray-finned fish group (Actinopterygii) because of its slow evolving and non-duplicated genome[85]. West Indian coelacanth (*Latimeria chalumnae*) and West African lungfish (*Protopterus annectens*) were used as a representative of the early-diverged lobe-finned fish group (Sarcopterygii)[86]. Chicken (*Gallus gallus*) and Chinese soft-shelled turtle (*Pelodiscus sinensis*) were included as archosaur representatives of the tetrapod group. In tetrapods, platypus (*Ornithorhynchus anatinus*), gray short-tailed opossum (*Monodelphis domestica*) and mouse (*Mus musculus*) were used as representatives of the three mammalian groups for the synteny analysis (monotreme, non-placental and placental mammals). Inshore hagfish (*Eptatretus burgeri*) was used as another representative of the cyclostomes (jawless vertebrates) in addition to sea lamprey in the phylogenetic analysis.

### Phylogenetic analysis of sea lamprey Cyp26 and Aldh1a proteins
Because protein alignments are more sensitive than DNA alignments and to avoid potential artefacts that come with the use of DNA sequences, we opted to perform protein alignments. Protein sequences corresponding to each gene were retrieved using both ENSEMBL and NCBI, focusing on the latest available version of the genome and/or the most complete available genome annotation. Amphioxus *Cyp26* and *Aldh1* gene families were used as outgroups for the analyses of *Cyp26* and *Aldh1a* complements, respectively. Protein sequences were aligned using the CLUSTALW protein alignment algorithm in MEGA11[87] and their evolutionary history was inferred using the Maximum Likelihood Tree method and WAG model[88] with 500 iterations for bootstrap testing, following Barry G. Hall's recommended parameters[89]. Protein accession numbers used to conduct the phylogenetic analyses are indicated in Supplementary Fig. 8.

### Synteny analysis
To help resolve the evolutionary history of the sea lamprey *Cyp26* and *Aldh1a* gene families with respect to their jawed vertebrate putative orthologs, we investigated the chromosomal organization of genes (i.e., synteny) located in the vicinity of *Cyp26* and *Aldh1a* genes in different vertebrate models using both ENSEMBL and NCBI Genome Data Viewer. Hagfish was not included in this synteny analysis, as the current version of the genome assembly lacked sufficient detail. The following gene families were selected for the synteny analysis: *Cyp26B1* synteny: *Znf638*, *Dysf*, *Exo6b*; *Cyp26A1/C1* synteny: *Myof*, *Exo6*; *Aldh1a2* synteny: *Aqp9*, *Myzap*, *Cgnl1*; *Aldh1a1* synteny: *Tmc1*, *Cyp1A/D*, *Anxa1*. mRNA accession numbers of *Cyp26* and *Aldh1a* used to conduct the synteny analyses in different vertebrate models are indicated in Supplementary Fig. 9.

### RNA-seq profile
RNAseq profiles corresponding to 1 to 5-day post fertilization (dpf)[50] as well as stages st18 to st24 (Tahara stages)[52,51] obtained from *P. marinus* dorsal neural tube tissues and/or whole embryos and accessible on the

KPetMar1 genome browser, were taken into consideration in the design of experiments described below investigating the gene expression of *Aldh1a* and *Cyp26* complements.

## Lamprey husbandry and embryo collection

Sea lamprey embryos were collected following the standard husbandry and culture protocols[90]. In brief, embryos were cultured in 0.05X Marc's Modified Ringers solution (MMR) embryo media (18 °C), staged according to Tahara stages, fixed in 1X MEMFA (1-part 10X MEMFA Salts, 1-part 37% formaldehyde, 8-parts $H_2O$) at appropriate developmental stages, dehydrated in 100% EtOH and stored at −20 °C for further experimentation.

## Colorimetric in situ hybridization (cISH)

RNA In Situ probes for *P. marinus Aldh1a* and *Cyp26* genes were designed based on gene sequences predicted by the Refseq model of KPetmar1 (*Aldh1a1/a2a*: XM_032949973.1; *Aldh1a1/a2b*: XM_032968791.1; *Cyp26A1*: XM_032952128.1; *Cyp26B1/C1a*: XM_032952126.1; *Cyp26B1/C1b*: XM_032950272.1). Because of the presence of repeat sequences as well as the unique profile of each gene sequence, individual probes were designed to target different gene locations. *Aldh1a1/a2a* RNA cISH probe was designed to target an internal region spanning from exon 5 to exon 9, based on the previously published mRNA sequence of '*aldh1a2*' (GenBank: FJ536260.1)[46]. For *Aldh1a1/a2b*, *Cyp26A1/B1a* and *Cyp26A1/B1b*, probes were designed to target the 3′ UTR sequence, while the RNA probe for *Cyp26A1* was designed to target a region overlapping the 5′UTR and the first exon of the gene. The following PCR primers were used to amplify probe templates:

*Aldh1a1/a2a* (597 bp, internal): F 5′- ACTTCACATTCACACGGC AC -3′; R: 5′-GCTTCTCGTCAATCTGTGGC-3′.

*Aldh1a1/a2b* (490 bp, 3′UTR): F 5′- TCCCGAAGAAGAGCTCGTAA-3′; R 5′- CCGCCGTAGAATTTAGAACG-3′.

*Cyp26A1* (423 bp, 5′UTR): F 5′- AAGAAAACCCCAACACAACG -3′; R 5′- AACTTCTCCCACAGCTTCCA-3′.

*Cyp26B1/C1a* (487 bp, 3′UTR): F 5′- GAGCGTCCTCTACAGCATCC -3′; R 5′- CACCATCTCTCCTCCGTCTC -3′.

*Cyp26B1/C1b* (402 bp, 3′UTR): F 5′- CCTGGGCTCAATAACGATGT -3′; R 5′- GCGCCACACTAAAAATCCAT -3′.

RNA cISH probe sequences for *P. marinus* hindbrain gene markers *otx*, *wnt1*, *krox20*, *kreisler*, *hoxβ1*, *hoxα2*, *hoxα3* and *hoxζ4* were described in previous work published from the lab[34,42]. Sequences were PCR amplified from *P. marinus* genomic DNA (gDNA) or st18-st26 embryonic cDNA using KOD Hot Start Master Mix (Millipore Sigma) with primers listed above. PCR products were cloned and sequenced from the PCR4Blunt-Topo Vector (ThermoFisher Scientific). Digoxigenin (DIG)-labeled RNA ISH probes were synthesized from selected clones following standard protocols and purified using the Megaclear Purification of Transcription Kit (Ambion). PCR products were cloned and sequenced from the PCR4Blunt-Topo Vector (ThermoFisher Scientific). DIG-labeled RNA ISH probes were synthesized from selected clones following standard protocols and purified using the Megaclear Purification of Transcription Kit (Ambion). In Situ Hybridization was performed on whole-mount embryos following the standard lamprey cISH protocol[90,91] optimized as recently described from our lab[92].

## Hybridization-chain reaction (HCR) fluorescent in situ hybridization (FISH)

For each gene, an HCR-FISH probe set targeting the coding sequence was custom-designed by Molecular Instruments with the number of probes varying between different sets. For each gene, the accession number, probe set size, amplifier/AlexaFluor dye used, and probe concentration are summarized in Supplementary Table 5. Because the sea lamprey embryo is highly opaque and auto-fluorescent, this HCR-FISH protocol needed to be adapted from the zebrafish HCRv3 FISH

protocol[60], with additional steps of bleaching and clearing of the tissue. First, sea lamprey embryos were gradually rehydrated at room temperature from 100% MeOH into 100% PBST (1X PBS + 0.1% Tween) in series of 5-minutes-long washes. Following this, embryos were then bleached using a freshly made solution (ingredients to be mixed in the following order: 5% Formamide, 0.5% SSC, 3% $H_2O_2$) then exposed under direct LED light (>10 K Lumen) for 2 h. This is followed by a few washes in 100% PBST. Embryos were kept at minimal density (about 5 embryos per tube) and were treated in 1 ml of proteinase K solution (20 µg/ml) for 8′, followed by a few washes in 100% PBST. Embryos were then re-fixed in 4% PFA for 20′ (shaking). The detection and amplification stage follows the zebrafish HCRv3 protocol[60]. The optimal probe input needs to be determined empirically for each probe set and target tissue and is further used to determine the volume of probe to be added to prepare a 2pmol probe/hybridization buffer solution (Supplementary Table 5).

## Imaging

For colorimetric in situ hybridization, embryos were gradually dehydrated in 100% MeOH post in situ hybridization and gradually cleared into 100% glycerol for imaging. Images were taken on a Leica MZ APO microscope using the Lumenera Infinity 3 camera at a magnification of 50X together with the Infinity Analyze software. Images were cropped and post-processed in a consistent way in Adobe Photoshop 2023 for color balance, brightness, and contrast. For HCR-FISH, embryos were cleared for 1 h in Optiprep (Stem Cell Technology). For dorsal imaging of the developing head, embryos were mounted dorsally on 0.5%-1% agarose blocks made with a customized inverted mold including a ~250 µm lane accommodating the head, and ~400 µm accommodating the yolk. Mounted embryos were carefully placed on a 35 mm MatTek glass bottom dish. For lateral imaging of the developing head, embryos were mounted laterally on a glass slide in Optiprep and positioned as flat as possible using layers of double-sided tape. Images were taken on a Nikon AT-AT inverted confocal microscope, using a 4X magnification for both lateral imaging of the developing head (st20 embryos) and dorsal imaging of the developing head (st21-st23.5 embryos). Images were analyzed and post processed for brightness and contrast in Fiji-ImageJ and Adobe Photoshop 2023, then cropped and assembled into montages in Adobe Illustrator 2023.

## Drug treatments

Sea lamprey embryos were treated with either 0.1% DiMethyl SulfOxide (DMSO) or with 10/50 µM of the pan-Aldh1 inhibitor DiEthylAmino-Benzaldehyde (DEAB) or with 10 µM of the Cyp26 inhibitor R115866 (Talarazole). Drug treatment was performed at gastrulation (st13/14) or neurulation (st16/17). Batches of embryos were incubated in a treatment solution made from diluting the drug solution 1000-fold into 0.05X MMR embryo media, cultured in the dark until fixation at st23. The treatment solution was replaced approximately every 36 h with care taken not to agitate embryos. Embryos were then fixed at appropriate developmental stages, dehydrated in 100% EtOH and stored at −20 °C for further experimentation.

## GFP-reporter experiments

A GFP reporter construct for zebrafish *hoxb3a* enhancer (*Drhoxb3a*-GFP) was made by PCR amplification of a 928 bp enhancer region of zebrafish *hoxb3a* followed by its cloning into the HLC (Hugo's Lamprey Construct) as previously described[34]. The reporter HLC plasmid (20ng µl-1) was digested by 0.5 U I-Sce enzyme in its adequate digestion buffer (NEB) for 30′ at 37 °C prior to injections. Sea lamprey embryos were injected at one-cell (st0-st2) with approximately 2 nl of the reporter construct and transient transgenesis was mediated by I-SceI meganuclease as previously described[34]. Approximately 600 embryos were injected with the reporter construct and approximately 200 embryos were treated by each drug. Only embryos displaying GFP

fluorescence at st23 were scored. Background GFP activity can be seen in the yolk and can be used as a marker of positively injected embryos. The ratio between the number of embryos displaying GFP fluorescence in the developing neural tube (i.e., positive reporter embryos) and the total number of GFP-positive embryos is listed in Fig. 6c. All representative phenotypes were imaged using Zeiss SteREO Discovery V12, fixed, dehydrated in 100% EtOH and stored at −20 °C for further experimentation. Images were cropped and post-processed in a consistent way in Adobe Photoshop 2023 for color balance, brightness, and contrast.

## CRISPR/Cas9 gene editing
Because of their important expression pattern during neural tube patterning, *Aldh1a1/a2a*, *Cyp26A1* and *Cyp26B1/C1a* were functionally disrupted using CRISPR/Cas9 gene editing. gRNAs were designed to target highly conserved enzymatic sites shared between sea lamprey and jawed vertebrates. Sea lamprey Aldh1a1/a2a protein sequence was aligned with jawed vertebrate Aldh1a2 protein sequences, jawed vertebrate Cyp26A1 proteins were aligned and sea lamprey Cyp26B1/C1a was aligned with jawed vertebrate Cyp26B1/C1 (Supplementary Figs. 1 and 2). Alignments were made using Clustal Omega Multiple Sequence Alignment tool. Conserved amino acid were highlighted in yellow on Snapgene, using a threshold of 95% for conservation. All gRNAs were designed using CRISPOR using KPetMar1 as a target genome and the standard -NGG PAM motif, and carefully selected to avoid off-target cutting. Two different gRNA were designed to target the highly conserved Cysteine and Glutamate residues of *Aldh1a1a2a* with the following sequences ($N_x$-*PAM*):

**g1**: AAGTTGATCCAGGAGGAGGC-*CGG*; **g2**: CCAGGGGGTGTTCTGGAACC-*AGG*.

Similarly, for each *Cyp26* gene, three different gRNA were designed to target the I helix, K helix and Heme Group, three highly conserved enzymatic sites of the Cyp26 family. gRNA sequences are as follows ($N_x$-*PAM*):

**Cyp26A1 gI**: GCCACGGAGCTCCTGTTCGG-*GGG*;
**Cyp26A1 gH1**: CGCTTCACCTACATCCCGTT-*CGG*;
**Cyp26A1 gH2**: TTCACCTACATCCCGTTCGG-*GGG*;
**Cyp26B1/C1a gK1**: CTACCTCGACTGCGTCGTCA-*AGG*;
**Cyp26B1/C1a gK2**: GCGTCGTCAAGGAAGTGCTG-*CGG*;
**Cyp26B1/C1a gH**: TTCAGCTACCTGCCGTTCGG-*CGG*.

A gRNA targeting the *Xenopus* tyrosinase gene was injected in each injected batch of embryos as a negative control (**gRNA neg**: GGCCCACTGCTCAGAAACCC)[93].

Following the IDT protocol 'Zebrafish embryo microinjection Ribonucleoprotein delivery using the AltR™ CRISPR-Cas9 System', contributed by Jeffrey Essner, PhD, we created a diluted 3 µM guide RNA (gRNA) solution by diluting a 100 µM Alt-R CRISPR-Cas9 RNA (crRNA) stock solution (IDT) and 100 µM Alt-R *Sp.* CRISPR-Cas9 tracrRNA stock solution (IDT) in nuclease-free water duplex buffer (IDT). A diluted 0.5 µg/µL Cas9 solution was made by diluting a Cas9 protein stock solution (10 µg/µl) (IDT) twenty-fold into Cas9 working buffer (20 mM HEPES; 150 mM KCl, pH 7.5, IDT). For each injection, ribonucleoprotein (RNP) complexes were assembled by combining 1 volume of gRNA and 1 volume of diluted Cas9 solutions into 2 volumes of injection buffer. In addition, 1 µl of 10% Lysinated Rhodamine-Dextran (LRD) tracer was added to each injection solution. The total volume of gRNA and Cas9 solution used varied between 1 to 2 µl according to the quality of the fertilization (i.e., number and quality of fertilized of embryos, death ratio etc.) with a more diluted solution injected in poorer fertilizations. When targeting individual genes (e.g., *Cyp26A1* alone), 2 or 3 gRNA were injected, 0.5 µl of each gRNA were used and mixed with 1 µl of Cas9 solution and 2 µl injection buffer. When targeting both *Cyp26A1* and *Cyp26B1/C1a* in the same embryo, we combined the following 4 gRNAs: **Cyp26A1 gH1**, **Cyp26A1 gI**,

**Cyp26B1/C1a gK1** and **Cyp26B1/C1a gH** and 0.25 µl of each gRNA were used and mixed with 1 µl of Cas9 solution and 2 µl injection buffer. Sea lamprey zygotes (st1-st2) were collected following external fertilization and immediately injected with approximately 3 nanoliters of RNP complex using a standard zebrafish micro-injection setup including injection needles, a microinjection manipulator and a microscope[94]. Embryo culture dishes were cleared for dead embryos the next day. Embryos showing fluorescence signal were cultured, fixed in MEMFA at appropriate developmental stages, dehydrated in 100% EtOH and stored at −20 °C for further experimentation.

## Genotyping of CRISPR embryos
Following cISH analysis of CRISPR embryos, representative embryos were genotyped to validate their mutant phenotypes. We genotyped Cr*Aldh1a/a2a* and Cr*Cyp26s* embryos in which both *Cyp26A1* and *Cyp26B1/C1a* were targeted. After rehydration, genomic DNA of CRISPR and control embryos was isolated by placing individual embryos in 45 µL of lysis buffer (10 mM Tris-HCl (pH8.0), 50 mM KCl, 0.3% Tween 20, 0.3% NP40, 4 mM EDTA), and incubated for 10' at 98 °C. Then 5 µl of Proteinase K (20 mg/ml) was added and the sample was incubated 55 °C for at least 24 h followed with a 10' incubation at 98 °C to denature the Proteinase K. The samples of genomic DNA were subsequently diluted in water (1:5). For each gene, primers were designed to amplify a genomic region spanning the targeted sites used in the CRISPR strategy (Supplementary Figs. 5 and 6). PCR amplification was performed using KOD Hot Start Master Mix (Millipore Sigma) with primers listed in Supplementary Table 6. PCR products were cloned using the PCR4Blunt-Topo Vector (ThermoFisher Scientific) and sequenced and the resulting sequences were aligned to the reference sequence from control CRISPR embryos. For each series of sequencing, the proportion of mutant loci is listed in Supplementary Figs. 5 and 6. We designed multiple sets of primers to amplify the region containing the K helix in the Cr*Cyp26s* embryos but were unsuccessful in amplifying this region. Further, sequencing results suggest that *Cyp26A1 gI* was not very efficient, as the majority of sequenced loci did not show a mutation in the I helix region (Supplementary Fig. 5).

## Reporting summary
Further information on research design is available in the Nature Portfolio Reporting Summary linked to this article.

## Data availability
The authors declare that all data supporting the findings of this study are available within the article and its supplementary information files or from the corresponding author upon reasonable request. Original data underlying this manuscript can be accessed from the Stowers Original Data Repository at http://www.stowers.org/research/publications/libpb-2405.

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

## Acknowledgements

We thank Brian Slaughter and Ruth Williams for providing guidance on generating and visualizing HCR data, Stephen Green, Megan Martik, and Tetsuto Miyashita for assistance with lamprey husbandry, and members of the Krumlauf laboratory for feedback on the experiments and manuscript. This study was conducted in accordance with the recommendations in the Guide for the Care and Use of Laboratory Animals of the NIH and protocols approved by the Institutional Animal Care and Use Committees of the California Institute of Technology (lamprey, MEB Protocol: #IA23-1436). This work was supported by a grant from the Stowers Institute for Medical Research to R.K. (grant #1001) and by a grant to M.E.B. (R35NS111564). This work was conducted to fulfill, in part, the requirements for A.M.H.B.'s thesis research as a Ph.D. student registered with the Open University, and we would like to thank the thesis committee members (Paul Trainor, Tatjana Piotrowski and Brian Slaughter) for their insightful comments and feedback.

## Author contributions

A.M.H.B., H.J.P., M.E.B and R.K. conceived this research program. A.M.H.B. and H.J.P. conducted the experiments and performed lamprey husbandry. A.J.P generated the RNAseq graph and J.A.M. contributed to optimizing the HCR protocol and generating HCR data. A.M.H.B., H.J.P., M.E.B. and R.K. analyzed the data, discussed the ideas and interpretations, and wrote the manuscript.

## Competing interests

The authors declare no competing interests.
