## [Peer Review File · Nature Communications]

Sea lamprey enlightens the origin of the coupling of retinoic acid signaling to vertebrate hindbrain segmentationREVIEWER COMMENTS

Reviewer #1 (Remarks to the Author):

In jawed vertebrates, RA signaling is intimately linked to the gene regulatory network (GRN) that regulates hindbrain segmentation within the developing central nervous system. This morphological and molecular segmentation is not observed in non-vertebrate chordates, and the conservation of these segmental features in jawless vertebrates is unclear. It therefore remains unresolved when this important developmental and regulatory mechanisms of central nervous system segmentation evolved within vertebrates - and this has important implications for understanding the evolution of central nervous system patterning and function.

In this manuscript, Bedois et al. report the findings of a beautiful and thorough study of the role of RA signaling in the lamprey, a jawless vertebrate developmental model system. Although they are unable to completely resolve the evolutionary history of gene families involved in the synthesis (Aldh1) and degradation (Cyp26) of retinoic acid within vertebrates - which is a common difficulty when building gene trees using sequences from living jawed and jawless vertebrates - they are able to recover clear Aldh1 and Cyp26 family members in the sea lamprey, *Petromyzon marinus*. They show that these genes are expressed in the expected time and place to be involved in hindbrain segmentation, and that their expression patterns correlated closely with known markers of jawed vertebrate hindbrain segmentation. And using both pharmacological and genome editing approaches, the authors show that RA signalling gain- or loss-of-function manipulations lead to clear and predictable molecular phenotypes relating the hindbrain segmentation.

I have no recommendations for revisions to this study. I think that the work is clear, thorough and well presented, and sets a very high standard for what can be done with this important model system. I strongly recommend publication.

Reviewer #2 (Remarks to the Author):

The general focus of the manuscript by Bedois et al is to clarify the origin of the mechanisms underlying hindbrain segmentation in vertebrates, focusing on the role of retinoic acid. This is a classical evo-devo question but (1) it is finally rarely comprehensively addressed, (2) hindbrain segmentation is an excellent model in view of the detailed data available in the mouse and zebrafish, (3) the lamprey is an organism of choice for this question and the authors make an excellent exploitation of relatively recent and powerful techniques in this species, such as HCR and CRISPR-cas9 gene editing. The experimental rationale is also extremely clearly explained, hypotheses and results are supported by an excellent iconography and the experimental data are of general of an excellent quality.

Although the data support broad conclusions, a number of points would improve the manuscript or add important precisions from an evolutionary standpoint.

1. Phylogenies.

In view of the usual difficulty to resolve orthology relationships between cyclostomes and gnathostomes, the authors take a relevant approach by considering all lamprey Cyp26 and Aldh1a paralogues, and by filtering those of interest based on transcriptomic data.

1.1. Although I doubt the major conclusions would be changed, the phylogenetic reconstructions (which yield groupings inconsistent with species phylogenies in gnathostomes) could be improved. For instance, the sampling of species, which is a bit biased with three mammals, could include a more representative sampling of gnathostomes, by adding another chondrichthyan (skate or elephant shark), a polypterid (such as the reedfish), the lungfish (which has a higher quality genome than the coelacanth), xenopus, and several appropriately selected archosaurs. Rooting with an amphioxus homologue would also be more relevant than using more distant gene families, such as Cyp51 or Aldh8.

1.2. In view of the difficulty to resolve relationships between cyclostome Cyp26B1/C1s and respective gnathostome Cyp26B1s and Cyp26C1s, can the authors exclude hidden paralogy between cyclostome and gnathostome Cyp26A1s? A scenario with differential paralogue losses between the two taxa from an ancestral syntenic Cyp26A1-Cyp26B1/C1 locus seems to me difficult to exclude and I am not sure that phylogeny resolves this point.

2. Expression analysis

The experimental data are of an excellent quality, with a relevant exploitation of HCR, a technique published for the first time in lamprey to my knowledge. More detailed conclusions as to the conservation of expression dynamics would be important for comprehensive comparisons.

1.1. Do the expression data support the conservation of successive (1) formation of subdivisions and (2) refinement/sharpening of boundaries similar to gnathostomes and can the timing of this dynamic, as inferred from expression data, in the lamprey be established?

1.2. In gnathostomes, timing is crucial in the elaboration of hindbrain segmentation, and the expression dynamic of Cyp26s reflects successive roles of RA in this multistep process. It is difficult from the data to have a comprehensive view of conservations of these individual steps with the lamprey, but also (for non-specialists of the system) across gnathostomes. A detailed scheme of the dynamic observed in the lamprey, including lamprey Cyp26A1 and both Cyp26B1/C1, and its comparison with the mouse and zebrafish would help to appreciate conservations and possible shuffling between paralogues. It would also provide a firm and clearer basis for very speculative points of the discussion about differences in gene intensity between the lamprey and zebrafish.

3. Functional analysis

Again, the experimental data are altogether of excellent quality and they support the main conclusions - involvement of RA in hindbrain segmentation and feedback controls in the signaling machinery - .However, a significant limitation is that they make it difficult to discriminate between successive roles of RA in hindbrain segmentation, and particularly, from the GRN angle taken from the authors, to isolate early, already documented posteriorizing roles of RA, from later ones.

Obviously, pharmacological treatments conducted at different time points and in defined developmental windows could clarify this question and considerably value the work. I understand that this may go

beyond the scope of this work, particularly in a species harboring a seasonal reproduction. In the absence of additional experimental data, the discussion should be expanded to clarify more specific hypotheses supported by the data as to the conservation of successive RA roles and of well documented genetic interactions (for instance between Krox20 and HoxB1), as established in established osteichthyan models.

Reviewer #3 (Remarks to the Author):

The present paper, entitled, Sea lamprey enlightens the origin of the coupling of retinoic acid signaling to vertebrate hindbrain segmentation, by Bedois et al., tried to show how retinoic acid (RA) is involved developmentally in the hypothetical gene regulatory network (GRN) that would function in the segmentation of the hindbrain in the lamprey, and preliminarily suggest the possibility that the RA is coupled to the hindbrain segmentation in the lamprey, which would have also been functioning in the common ancestor of the entire vertebrates. To draw these conclusions, the authors analysed not only expressions of hindbrain-segments (rhombomeres) marker genes as well as the genes involved in RA synthesis and degradation, but also resort to functional analyses Aldh1as and Cyp26s on lamprey embryos. As the markers of hindbrain segments, lamprey homologues of Krox20, kreisler, and some Hox genes (known to be expressed in the mouse hindbrain region) are employed. There are no problems in the molecular phylogenetic analyses of genes. Experiments are carefully done, and gene expressions are clear and trustworthy in the figures. Text is clearly written as well.

Major comments:

I believe this paper will potentially make an important contribution to the evolutionary developmental understanding of the vertebrate embryogenesis especially in the context of GRN evolution. Most puzzling, however, is that there are no data or figures of hindbrain segmentation shown anywhere in the paper. Apparently, Krox20 expression is regarded as a marker to show the presence of r3 and r5 in the lamprey, which, however, is only expected from the expression pattern of the orthologue gene in mouse embryos (I do not understand why the authors neglect the previous papers describing the expression of Krox20 in the lamprey). Of course, gene expression itself is not equal to neurepithelial segmentation, which should be discussed based on anatomical examination, the formation of neurepithelial boundaries and its mechanisms.

More problematic is the expression of HoxPG1-4 genes in the lamprey. In the present paper, Hox β 1 is shown to correspond to r4, but it is not tested whether all the other Hox genes have expression boundaries corresponding to segment boundaries of the hindbrain. This should be shown more clearly at the histological level with the photos of rhombomere boundaries that should be observable in older embryos. This is a very important point since, in the lamprey, it has already been reported that HoxPG3 gene has an expression boundary in the mid of r4, corresponding to the boundary between trigeminal and facial motoneuronal populations (Murakami et al., 2004).

In the above connection, statement from L 87, “It was previously postulated that the lamprey hindbrain employed a rudimentary form of segmentation, which is only partially coupled to Hox expression and does not involve the action of RA signaling, suggesting roles for RA in hindbrain segmentation arose later in vertebrate evolution^{39, 40}.” is a misleading interpretation of citations.

The term segmentation primarily refers to division of tissues or structures, however, evolutionary developmental context, it often carries connotations of serial homologues or metamerical organization in animal body plan, for which the term ‘metamerical segmentation’ would be more preferable. The authors appear to assume RA-mediated Hox regulation behind hindbrain segmentation as a classical developmental network, but from a comparative embryological perspective, Hox expressions are more likely coupled to branchial motoneurons’ distribution along the AP axis (rhombomeres are apparently more likely coupled to the segmental distribution of reticulospinal neurons – Murakami et al., 2004). It is quite unfortunate that this evolutionary morphological evidence, apparently suggesting an evolutionary polarity in the coupling the Hox regulation and segmentation, is not taken into account in the manuscript. In other words, this paper only tries to identify synplesiomorphies, and neglect possible apomorphies either in cyclostomes or gnathostomes, which is not very different from the 19th century’s archetype theory.

For the reason stated above, the title of the paper, “...origin of the coupling of retinoic acid signaling to vertebrate hindbrain segmentation” sounds to be an overstatement, and should be changed more appropriately like “...origin of the coupling of retinoic acid signaling to regulation of Krox20 in the vertebrate hindbrain”. Also, the paper should be extensively revised, focusing on the mechanism of gene expressions. At this point, I cannot recommend this paper for publication, however after the revision, as I stated, it will make an important contribution to our understanding of GRN evolution behind the vertebrate body plan, in which case, I am afraid I may not be a good candidate as the referee.

Minor comments:

L. 68 "... lays ground for patterning neuronal differentiation..." seems to deify the existence of rhombomeres more than necessary, and even contradicts the fact that they emerged secondarily in the chordate lineage.

L. 98: “This conservation implies that hindbrain segmentation...”. Is this sentence necessary? I believe this has long been known already.

L. 213: “... suggesting that Cyp26B1/C1a is also coupled...”. Overstatement.

L. 240: During early hindbrain segmentation. Unclear and data not shown. Is stage 21 really the stage when the earliest rhombomere appears in the lamprey?

L. 251-252: Too speculative.

L. 425: Hox patterning. I do not understand what it means.

L. 534: It may be important to realize that the entire history of whole genome duplication may not be entirely shared between gnathostome and cyclostome lineages. This might deepen the discussion here.

Detailed list of changes to manuscript and response to reviewers' comments.

Reviewers Comments

Reviewer #1 (Remarks to the Author):

In jawed vertebrates, RA signaling is intimately linked to the gene regulatory network (GRN) that regulates hindbrain segmentation within the developing central nervous system. This morphological and molecular segmentation is not observed in non-vertebrate chordates, and the conservation of these segmental features in jawless vertebrates is unclear. It therefore remains unresolved when this important developmental and regulatory mechanisms of central nervous system segmentation evolved within vertebrates - and this has important implications for understanding the evolution of central nervous system patterning and function.

In this manuscript, Bedois et al. report the findings of a beautiful and thorough study of the role of RA signaling in the lamprey, a jawless vertebrate developmental model system. Although they are unable to completely resolve the evolutionary history of gene families involved in the synthesis (Aldh1) and degradation (Cyp26) of retinoic acid within vertebrates - which is a common difficulty when building gene trees using sequences from living jawed and jawless vertebrates - they are able to recover clear Aldh1 and Cyp26 family members in the sea lamprey, *Petromyzon marinus*. They show that these genes are expressed in the expected time and place to be involved in hindbrain segmentation, and that their expression patterns correlated closely with known markers of jawed vertebrate hindbrain segmentation. And using both pharmacological and genome editing approaches, the authors show that RA signalling gain- or loss-of-function manipulations lead to clear and predictable molecular phenotypes relating the hindbrain segmentation.

I have no recommendations for revisions to this study. I think that the work is clear, thorough and well presented, and sets a very high standard for what can be done with this important model system. I strongly recommend publication.

We are glad that this reviewer found our work thorough, clear and important.

Reviewer #2 (Remarks to the Author):

The general focus of the manuscript by Bedois et al is to clarify the origin of the mechanisms underlying hindbrain segmentation in vertebrates, focusing on the role of retinoic acid. This is a classical evo-devo question but (1) it is finally rarely comprehensively addressed, (2) hindbrain segmentation is an excellent model in view of the detailed data available in the mouse and zebrafish, (3) the lamprey is an organism of choice for this question and the authors make an excellent exploitation of relatively recent and powerful techniques in this species, such as HCR and CRISPR-cas9 gene editing. The experimental rationale is also extremely clearly explained, hypotheses and results are supported by an excellent iconography and the experimental data are of general of an excellent quality.

Although the data support broad conclusions, a number of points would improve the manuscript or add important precisions from an evolutionary standpoint.

1. Phylogenies.

In view of the usual difficulty to resolve orthology relationships between cyclostomes and gnathostomes, the authors take a relevant approach by considering all lamprey Cyp26 and Aldh1a paralogues, and by filtering those of interest based on transcriptomic data.

1.1.) Although I doubt the major conclusions would be changed, the phylogenetic reconstructions (which yield groupings inconsistent with species phylogenies in gnathostomes) could be improved. For instance, the sampling of species, which is a bit biased with three mammals, could include a more representative sampling of gnathostomes, by adding another chondrichthyan (skate or elephant shark), a polypterid (such as the reedfish), the lungfish (which has a higher quality genome than the coelacanth), xenopus, and several appropriately selected archosaurs. Rooting with an amphioxus homologue would also be more relevant than using more distant gene families, such as *Cyp51* or *Aldh8*.

This is a good point and we have followed their advice generating two new phylogenetic trees that integrate the suggested changes, (i.e., adding reedfish, lungfish, thorny skate, chicken and a turtle as archosaurs, removing platypus and opossum and keeping only mouse as a mammalian representative). We also rooted the trees with amphioxus homologues. Accession numbers of proteins used to generate these new trees have been added to Supplementary Figure 8. As the reviewer predicted, our conclusions remain unchanged regarding the possible evolution of the Aldh and Cyp26 genes in vertebrates, but we're hoping that these new trees are more accurate. Hence, we have replaced our original trees with these new trees in a revised Figure2a.

1.2.) In view of the difficulty to resolve relationships between cyclostome Cyp26B1/C1s and respective gnathostome Cyp26B1s and Cyp26C1s, can the authors exclude hidden paralogy between cyclostome and gnathostome Cyp26A1s? A scenario with differential paralogue losses between the two taxa from an ancestral syntenic Cyp26A1-Cyp26B1/C1 locus seems to me difficult to exclude and I am not sure that phylogeny resolves this point.

*It is true that hidden paralogy between cyclostome and gnathostome Cyp26A1s cannot be excluded based on the phylogenetic analysis. Thus, the cyclostome and gnathostome Cyp26A1s may not be direct orthologues. Hence, we added the following sentences in the results section (**Identification and phylogenetic analysis of sea lamprey Cyp26 and Aldh1a gene families.**), at the end of the first paragraph describing the evolution of the Cyp26 family in vertebrates:*

“In addition, hidden paralogy between cyclostome and gnathostome Cyp26A1s cannot be excluded based on the phylogenetic analysis. Thus, it is possible that the cyclostome and gnathostome Cyp26A1s may not be direct orthologues.”

2. Expression analysis

The experimental data are of an excellent quality, with a relevant exploitation of HCR, a technique published for the first time in lamprey to my knowledge. More detailed conclusions as to the conservation of expression dynamics would be important for comprehensive comparisons.

This is an excellent point. We were originally abbreviated in our descriptions of the expression patterns to reduce length, but we agree providing slightly more detailed conclusions based on the comparisons is important. Towards this end we have made the modifications described below.

2.1). Do the expression data support the conservation of successive (1) formation of subdivisions and (2) refinement/sharpening of boundaries similar to gnathostomes and can the timing of this dynamic, as inferred from expression data, in the lamprey be established?

*These are very good points. To address and clarify the issue we have added the following text to the end of the section on endogenous expression patterns (**Expression of Cyp26s and Aldh1a genes is linked to lamprey hindbrain segmentation.**)*

“The expression dynamics of lamprey Cyp26 genes appear to support conservation of similar phases to those seen in gnathostomes, when considered in relation to morphological events and the expression of other patterning genes. Anterior Cyp26A1 expression between gastrulation and neurulation (st16-20) coincides with the onset of expression of genes such as HoxPG1, vHnf1, and kreisler, which in gnathostomes are involved in formation of molecular subdivisions that prefigure rhombomeres. Later segmental expression of Cyp26B1/C1a at st21-23 in lamprey coincides with the period in which rhombomeres become visible morphologically, and when krox20, kreisler and Hox genes are segmentally expressed. Thus, it appears that the successive stages in the formation of subdivisions and sharpening or refining these boundaries seen in gnathostomes such as zebrafish, are also reflected in the expression of the lamprey Cyp26 genes during hindbrain development.”

It is also important to mention that we don't have good markers for rhombomere boundaries in lamprey and therefore we can't say when they appear and whether there are mechanisms in place preventing mixing of adjacent cells. In addition, HCR isn't the method of choice to address the refinement of boundaries more precisely than by looking at successive stages. We present three stages in our HCR panel (Figure 4), representing successive steps in the formation of segments but we also did some HCR at intermediate stages (st20.5 and st21.5) but didn't see any difference in expression compared to st20 and st21. To address the refinement of boundaries in more detail,

in future we would need to look at it in living embryos but the very long developmental timing of lamprey make it very difficult to address this point.

2.2). In gnathostomes, timing is crucial in the elaboration of hindbrain segmentation, and the expression dynamic of *Cyp26s* reflects successive roles of RA in this multistep process. It is difficult from the data to have a comprehensive view of conservations of these individual steps with the lamprey, but also (for non-specialists of the system) across gnathostomes. A detailed scheme of the dynamic observed in the lamprey, including lamprey *Cyp26A1* and both *Cyp26B1/C1*, and its comparison with the mouse and zebrafish would help to appreciate conservations and possible shuffling between paralogues. It would also provide a firm and clearer basis for very speculative points of the discussion about differences in gene intensity between the lamprey and zebrafish.

This is an excellent point. To help address this issue, we generated a simplified schematic summarizing the similarities and differences in the dynamic of expression of Cyp26 genes in mouse, zebrafish and lamprey embryos, which is now Supplementary Figure 10. This notably illustrates the similarities in Cyp26A1 expression territories in early development (gastrulation and early neurulation) and shows the differences in expression of zebrafish Cyp26b1/Cyp26c1, mouse Cyp26B1/Cyp26C1 and lamprey Cyp26B1/C1a, in terms of their relative level of expression in various rhombomeres, during hindbrain segmentation. This schematic has been added as Supplementary Figure 10, and we made modifications in the second paragraph of the discussion to refer to it.

3. Functional analysis

Again, the experimental data are altogether of excellent quality, and they support the main conclusions -involvement of RA in hindbrain segmentation and feedback controls in the signaling machinery. However, a significant limitation is that they make it difficult to discriminate between successive roles of RA in hindbrain segmentation, and particularly, from the GRN angle taken from the authors, to isolate early, already documented posteriorizing roles of RA, from later ones.

Obviously, pharmacological treatments conducted at different time points and in defined developmental windows could clarify this question and considerably add value to the work. I understand that this may go beyond the scope of this work, particularly in a species harboring a seasonal reproduction. In the absence of additional experimental data, the discussion should be expanded to clarify more specific hypotheses supported by the data as to the conservation of successive RA roles and of well documented genetic interactions (for instance between *Krox20* and *HoxB1*), as established in established osteichthyan models.

We agree that CRISPR approaches to create deletion mutants reveal initial or early roles for RA but make it difficult to uncover and understand later or successive roles for RA. Unfortunately, conditional approaches for temporal deletion of genes in lamprey are not yet available. As the reviewer suggests pharmacological treatments at different developmental time points could help to clarify or address this issue.

*We have conducted some of these types of experiments as a part of the process of trying to determine the optimal timing and concentrations of the pharmacological inhibitors to assess their effects on hindbrain segmentation. For example, we conducted a series of treatments using 10 μ M or 50 μ M of DEAB starting during early gastrulation (st13) or neurulation (st17). We found that treating with 10 μ M DEAB at early gastrulation (st13) induces some mis-patterning of the hindbrain, as evidenced by the loss of *hoxz4* expression and lower expression of some other markers, while treating later at early neurulation (st17) has weaker to no effect (see new Supplementary Figure 7). In contrast, treating with 50 μ M clearly shows severe mis-patterning of the hindbrain when added at both gastrulation (st13) and early neurulation (st17) stages. We conclude from these experiments that the influence of RA on hindbrain patterning and later segmentation starts around the beginning of gastrulation, and persists in later stages, suggestive of successive roles for RA. This is further supported by evidence from ISH of expression of individual *Cyp26A1*, *Cyp26B1/C1a* and *Aldh1a1/a2a* at this stage (Figure 3).*

We feel it would be useful to include these additional experiments to begin to address the point raised by the reviewer on successive roles for RA. In the revised manuscript we replaced the original Supplementary Figure 7, which showed a series of ISH in embryos treated with 10 μ M of DEAB during st13-st23, by a new panel (Supplementary Figure 7a) that includes the experiments described above, with two concentrations of DEAB and different stages, together with a summary cartoon of the results (Supplementary Figure 7b). We also added the following text to the discussion referring to the timing of the DEAB effects on hindbrain segmentation.

“This study reveals a key role for RA signaling in influencing hindbrain segmentation, starting during the early steps of gastrulation (st13) but what about later stages? We have found that lamprey embryos treated with 50 μ M DEAB at later time points, early neurulation stages (st17), also display severe defects in hindbrain patterning (Supplementary Figure 7). This suggests that as in gnathostomes, RA appears to have multiple and successive roles in regulating the hindbrain GRN in lamprey. It is also possible that RA might play additional roles in developmental processes happening after hindbrain segmentation has reached completion, including neural crest patterning and neurogenesis.”

Reviewer #3 (Remarks to the Author):

The present paper, entitled, Sea lamprey enlightens the origin of the coupling of retinoic acid signaling to vertebrate hindbrain segmentation, by Bedois et al., tried to show how retinoic acid (RA) is involved developmentally in the hypothetical gene regulatory network (GRN) that would function in the segmentation of the hindbrain in the lamprey, and preliminarily suggest the possibility that the RA is coupled to the hindbrain segmentation in the lamprey, which would have also been functioning in the common ancestor of the entire vertebrates. To draw these conclusions, the authors analyzed not only expressions of hindbrain-segments (rhombomeres) marker genes as well as the genes involved in RA synthesis and degradation, but also resort to functional analyses Aldh1as and Cyp26s on lamprey embryos. As the markers of hindbrain segments, lamprey homologues of Krox20, kreisler, and some Hox genes (known to be expressed in the mouse hindbrain region) are employed. There are no problems in the molecular phylogenetic analyses of genes. Experiments are carefully done, and gene expressions are clear and trustworthy in the figures. Text is clearly written as well.

Major comments:

I believe this paper will potentially make an important contribution to the evolutionary developmental understanding of the vertebrate embryogenesis especially in the context of GRN evolution. Most puzzling, however, is that there are no data or figures of hindbrain segmentation shown anywhere in the paper. Apparently, Krox20 expression is regarded as a marker to show the presence of r3 and r5 in the lamprey, which, however, is only expected from the expression pattern of the orthologous gene in mouse embryos (I do not understand why the authors neglect the previous papers describing the expression of Krox20 in the lamprey). Of course, gene expression itself is not equal to neuroepithelial segmentation, which should be discussed based on anatomical examination, the formation of neuroepithelial boundaries and its mechanisms.

Given that rhombomeres are faint and relatively difficult to visualize in lamprey, we used molecular markers such as krox20 and kreisler to mark rhombomeres in our study. These markers provide a readout of the conserved molecular regionalization that demarcates neuroepithelial segments in gnathostomes. Previous studies, including our own, have clearly shown that krox20 expression marks r3 and r5 in lamprey (Murakami et al. 2004; Jimenez-Guri et al. 2011; Parker et al. 2014; Parker et al. 2019a). In lamprey embryos, transient rhombomeric segmentation has been described by analyses of morphology (by electron microscopy (Horigome et al. 1999)), acetylated tubulin immunostaining and histology (Kuratani et al. 1998), and by the segmental expression of marker genes including lamprey homologues of krox20 (Murakami et al. 2004), vhnf1 (Jimenez-Guri et al. 2011), ephA4 (Murakami et al. 2004) and kreisler (Parker et al. 2014). Taken together, these studies demonstrate that rhombomeric boundaries are relatively faint and are only visible for a short period during development (st22-24), compared with those in gnathostomes. Importantly, the cranial nerve roots emanate from specific rhombomeres, such as the facial root from r4, and the otic vesicle lies adjacent to r4 (Kuratani et al. 1998; Horigome et al. 1999). Using these landmarks, the expression domains of lamprey krox20 clearly correspond to r3 and r5 (Murakami et al. 2004). We apologize for neglecting to include the entirety of previous papers describing lamprey krox20 expression; we have added some citations at relevant places in the manuscript to rectify this.

*We have added the following sentences in the results section (**Pharmacological modulation of endogenous RA levels impacts hindbrain segmentation.**) to justify our approach, when introducing the use of molecular markers:*

“Given that rhombomeres are faint and relatively difficult to visualize in lamprey, we chose to use molecular markers such as krox20 and kreisler and several Hox genes to mark rhombomeres in our study. These markers have been used in previous studies (Murakami et al. 2004; Jimenez-Guri et al. 2011; Parker et al. 2014; Parker et al. 2019a) and provide a readout of the molecular regionalization that demarcates neuroepithelial segments.”

More problematic is the expression of HoxPG1-4 genes in the lamprey. In the present paper, Hoxβ1 is shown to correspond to r4, but it is not tested whether all the other Hox genes have expression boundaries corresponding to segment boundaries of the hindbrain. This should be shown more clearly at the histological level with the photos of rhombomere boundaries that should be observable in older embryos. This is a very important point since, in the lamprey, it has already been reported that HoxPG3 gene has an expression boundary in the mid of r4, corresponding to the boundary between trigeminal and facial motoneuronal populations (Murakami et al., 2004).

We agree that testing the correspondence between hox expression and segment boundaries in the lamprey hindbrain is important. We have previously tested this at st23-24 by double in situ hybridization of hox genes with the segmental markers krox20 (r3r5) or kreisler (r5) (Parker et al. 2014; Parker et al. 2019a). This revealed that many HoxPG1-4 genes, including hoxa3, show segment-specific expression domains at st21-24 that are similar to those seen in gnathostomes. It is also notable that the hagfish hindbrain is transiently segmented into rhombomeres and that hagfish hox expression boundaries correspond with specific rhombomeres in a broadly similar manner to lamprey and gnathostomes (Pascual-Anaya et al., 2018). This further supports an ancestral alignment between hox gene expression and rhombomeres in vertebrates. Some lamprey hox genes subsequently depart from segmental restriction at later stages. For instance, hoxa3 is initially expressed up to the r4/r5 border at st22-23 but then spreads anteriorly into specific neuronal populations within r4 (Parker et al 2014 & 2019a). Thus, lamprey hox expression is transiently connected with specific rhombomeres but then some genes escape segmental restriction. Murakami et al. (2004) characterized expression of the homologue of hoxa3 in Japanese/arctic lamprey (Lethenteron Japonicum/Camtschaticum). However, they focused on st26 and not earlier stages, so it is possible that earlier segmental domains were missed (Murakami et al., 2004). This dynamic hoxa3 expression in sea lamprey is interesting and may indicate early segment-specific roles and later segmentally independent roles in specifying certain neuronal populations. This is an interesting area for future functional investigation of lamprey hox genes but we feel it is beyond the scope of the current study.

Since this point on dynamics of hoxa3 expression also relates to the reviewer's later point regarding apomorphies in cyclostomes or gnathostomes, and also to point 3 of reviewer 2 on successive roles of RA, we have added the following text to the discussion:

“While our data supports a shared vertebrate ancestry of the influence of RA in the genetic program for hindbrain segmentation, we do not exclude the importance of derived features in hindbrain development between gnathostome and cyclostome lineages. Indeed, while lamprey hox expression is transiently connected with specific rhombomeres, some hox genes subsequently escape segmental restriction at later stages. For example, hoxa3 is initially expressed up to the r4/r5 border at st22-23 but then spreads anteriorly in specific neuronal populations within r4. This appears to be different from the situation in gnathostomes, where such escape of hox expression from segmentation has not been described. Thus, based on current evidence in lamprey, it appears that Hox genes are initially coupled with the RA-dependent formation and early patterning of rhombomeres but subsequently Hox expression domains are not all maintained in segmental register, which may alter later roles in branchiomotor neuron positioning.

This study reveals a key role for RA signaling in influencing hindbrain segmentation, starting during the early steps of gastrulation (st13) but what about later stages? We have found that lamprey embryos treated with 50 μ M DEAB at later time points, early neurulation stages (st17), also display severe defects in hindbrain patterning (Supplementary Figure 7). This suggests that as in gnathostomes, RA appears have multiple and successive roles in regulating the hindbrain GRN in lamprey. It is also possible that RA might play additional roles in developmental processes happening after hindbrain segmentation has reached completion, including neural crest patterning and neurogenesis.”

In the above connection, statement from L 87, “It was previously postulated that the lamprey hindbrain employed a rudimentary form of segmentation, which is only partially coupled to Hox expression and does not involve the action of RA signaling, suggesting roles for RA in hindbrain segmentation arose later in vertebrate evolution^{39, 40.}” is a misleading interpretation of citations.

We made the following change in the text referred to by the reviewer. It now states:

“It was previously postulated that lamprey hindbrain segmentation is only partially coupled to Hox expression, and that RA signaling influences Hox-dependent branchiomotor neuron specification but not hindbrain segmentation itself (Murakami 2004, 2005). This implies that roles for RA in hindbrain segmentation may have arisen later in vertebrate evolution.”

The term segmentation primarily refers to division of tissues or structures, however, in evolutionary developmental context, it often carries connotations of serial homologues or metameric organization in animal body plan, for which the term ‘metameric segmentation’ would be more preferable. The authors appear to assume RA-mediated Hox regulation behind hindbrain segmentation as a classical developmental network, but from a comparative embryological perspective, Hox expressions are more likely coupled to branchial motoneurons’ distribution along the AP axis (rhombomeres are apparently more likely coupled to the segmental distribution of reticulospinal neurons – Murakami et al., 2004). It is quite unfortunate that this evolutionary morphological evidence, apparently suggesting an evolutionary polarity in the coupling the Hox regulation and segmentation, is not taken into account in the manuscript. In other words, this paper only tries to identify synplesiomorphies, and neglect possible apomorphies either in cyclostomes or gnathostomes, which is not very different from the 19th century’s archetype theory.

Regarding evolutionary morphology, we do not exclude the importance of apomorphies in hindbrain development between gnathostome and cyclostome lineages or that RA-mediated Hox regulation was likely coupled to motor neurons along the AP axis in the ancestral chordate. RA and Hox genes also appear to have deeply conserved roles as part of an ancient head to tail axial patterning system in chordate evolution. We feel it is important to clarify that we consider our work and that of Murakami et al. 2004 to be complementary studies that focus on different stages of hindbrain development. While our data shows that RA does influence the GRN for hindbrain segmentation in lamprey and supports synplesiomorphy in the genetic program involved in hindbrain segmentation, each group may also have specific features. An example may be the apparent escape from segmental restriction of hox expression and branchiomotor neuron distribution at later stages in lamprey embryos (which is a key finding from Murakami et al. 2004). This appears to be different from the situation in gnathostomes, where such escape of hox expression from segmentation has not been described. Thus, based on current evidence in lamprey, it is tempting to speculate that Hox genes are initially coupled with the RA-dependent formation and early patterning of rhombomeres but subsequently Hox expression domains are not all maintained in segmental register, which may alter later roles in BMN positioning. We do not explore or test that

hypothesis in this study but our data also points to some important differences in the expression and potential role of Cyp26s between agnathans and gnathostomes. Investigating whether there are dynamic temporal roles for these genes in the lamprey hindbrain will be important and understanding how their arise due to differences in wiring of the GRN will give important insights into evolution of this vertebrate trait, but this is beyond the scope of the current manuscript.

For the reason stated above, the title of the paper, "...origin of the coupling of retinoic acid signaling to vertebrate hindbrain segmentation" sounds to be an overstatement, and should be changed more appropriately like "...origin of the coupling of retinoic acid signaling to regulation of Krox20 in the vertebrate hindbrain".

The suggested title is too reductionist and does not reflect our results on virtually all of the molecular components of the hindbrain GRN evaluated in the study. We strongly feel our original title appropriately illustrates what the study is about and its conclusions.

Also, the paper should be extensively revised, focusing on the mechanism of gene expressions.

Now that we have clearly established that RA plays an important regulatory role in the lamprey hindbrain GRN it will be important in future to characterize the cis-regulatory circuitry. The goal of this study was to investigate if RA signaling was present and played a potential role in hindbrain segmentation in sea lamprey as a jawless vertebrate model. Past work suggested that it was not important in the lamprey hindbrain for segmentation, but we have firmly established its key role in the early lamprey hindbrain. Investigating the molecular mechanisms regulating gene expression of Cyp26 in jawless vertebrates is indeed an important point but is beyond the scope of this study. However, it is worth noting that we have begun to investigate the basis of Cyp26 regulation by looking into the presence of RAREs in the vicinity of Cyp26A1 to investigate whether the RA-regulation of Cyp26A1 expression is shared in all vertebrates. We are hoping that this investigation will result in regulatory insights into how RA signaling is directly wired into the hindbrain GRN in lamprey and will allow us to compare the regulatory circuitry in more detail.

At this point, I cannot recommend this paper for publication, however after the revision, as I stated, it will make an important contribution to our understanding of GRN evolution behind the vertebrate body plan, in which case, I am afraid I may not be a good candidate as the referee.

Minor comments:

L. 68 "... lays ground for patterning neuronal differentiation..." seems to deify the existence of rhombomeres more than necessary, and even contradicts the fact that they emerged secondarily in the chordate lineage.

We intended this to refer to regional patterning and have modified the sentence to read:

"During early embryogenesis, the hindbrain is transiently organized into segments (rhombomeres), which lays down a ground plan for regional patterning of neural differentiation, circuit formation and head development."

L. 98: “This conservation implies that hindbrain segmentation...”. Is this sentence necessary? I believe this has long been known already.

We think this sentence is absolutely necessary and fits well after the previous sentence as it clearly explains what we refer to when talking about segmentation in this study, mostly at the molecular level, as compared to other studies who may refer more to the morphological aspect of segmentation (Murakami, 2004).

L. 213: “... suggesting that Cyp26B1/C1a is also coupled...”. Overstatement.

The full sentence is “This dynamic pattern is reminiscent of the expression of Cyp26B1 and Cyp26C1 in specific rhombomeres of zebrafish and mouse embryos suggesting that Cyp26B1/C1a is also coupled to segmentation in lamprey.”

What we meant by this sentence is that based on the expression pattern of these genes in specific rhombomeres in mouse and zebrafish and the role of Cyp26s in modulating the levels of RA, therefore being coupled to the process of segmentation, we suggest here the possibility that lamprey Cyp26B1/C1a could also be coupled to segmentation, since it is expressed in specific segments.

We propose the following change: “Cyp26B1/C1a could also be coupled to segmentation.”

L. 240: During early hindbrain segmentation. Unclear and data not shown. Is stage 21 really the stage when the earliest rhombomere appears in the lamprey?

*The data can be seen in the HCR panel, st21 in Figure 4. Lamprey rhombomeres are faint but have been described as morphologically visible from st22. However, gene expression clearly shows defined and abutting stripes of *krox20*, *hoxβ1*, *hoxa2* etc. at st21, which could technically be called ‘pre-rhombomeres’, hence our use of the term ‘early hindbrain segmentation’. We have also showed the dynamics of expression of segmental markers in lamprey hindbrain segmentation in several previous studies (Parker et al 2014 and 2019a).*

L. 251-252: Too speculative.

*The full sentence is “Cyp26B1/C1a and *krox20*, a marker of r3 and r5, which is initially expressed in r3 and then in r3/r5 in lamprey and other vertebrates^{34, 61, 62}.” The lamprey data is shown in the cited reference (Parker et al, 2014) as well as in the earlier publication Jimenez-Guri et al. 2011. In addition to these published results, the HCR panel of Figure 4 of this study, at st20, clearly illustrates this as well.*

L. 425: Hox patterning. I do not understand what it means.

We have removed that part of the sentence for simplification.

L. 534: It may be important to realize that the entire history of whole genome duplication may not be entirely shared between gnathostome and cyclostome lineages. This might deepen the discussion here.

Relevant to this point we added the following sentence in the discussion,

*“Furthermore, additional independent rounds of whole genome duplication have recently been inferred in each lineage after their divergence (Simakov et al 2020). These may have provided even further scope for functional regulatory divergence of *cyp26* genes in each lineage.”*

REVIEWERS' COMMENTS

Reviewer #2 (Remarks to the Author):

The authors have significantly expanded experimental data, including functional analyses, and their additions fully answer my questions. The final manuscript provides an important contribution to the origin of segmentation in the vertebrate hindbrain and it opens interesting perspectives to further explore its diversifications in a major model organism in vertebrate evolutionary developmental biology. I noticed a very minor typo in Figure 2a: *Petromyzon* instead of *Petromizon*, plus species names in italics in the top left of the figure.